# Preference Learning of Latent Decision Utilities with a Human-like Model of Preferential Choice

**Sebastiaan De Peuter**
Aalto University
sebastiaan.depeuter@aalto.fi

**Shibei Zhu**
Aalto University
shibei.zhu@aalto.fi

**Yujia Guo**
Aalto University
yujia.guo@aalto.fi

**Andrew Howes**
University of Exeter
andrew.howes@exeter.ac.uk

**Samuel Kaski**
Aalto University
University of Manchester
samuel.kaski@aalto.fi

## Abstract

Preference learning methods make use of models of human choice in order to infer the latent utilities that underlie human behavior. However, accurate modeling of human choice behavior is challenging due to a range of context effects that arise from how humans contrast and evaluate options. Cognitive science has proposed several models that capture these intricacies but, due to their intractable nature, work on preference learning has, in practice, had to rely on tractable but simplified variants of the well-known Bradley-Terry model. In this paper, we take one state-of-the-art intractable cognitive model and propose a tractable surrogate that is suitable for deployment in preference learning. We then introduce a mechanism for fitting the surrogate to human data and extend it to account for data that cannot be explained by the original cognitive model. We demonstrate on large-scale human data that this model produces significantly better inferences on static and actively elicited data than existing Bradley-Terry variants. We further show in simulation that when using this model for preference learning, we can significantly improve utility in a range of real-world tasks.

## 1 Introduction

AI systems need exact descriptions of tasks to be performed. However, humans find more complex tasks hard to describe. In response, preference learning has emerged as one way to learn from human feedback. It has been used to teach AI systems a variety of tasks from how to hand objects to humans to how to play Atari games [1–4]. More recently, human feedback has been used to train large language models to summarize text [5], answer questions in natural language [6], and to train deep generative models to generate realistic medical images [7].

When learning from human feedback, it is generally assumed that some latent utility function $f$ guides an individual's behavior, but that the individual cannot describe this function to the machine. Thus, *preference queries* are used to elicit information about $f$ from the user. A preference query gives a user a set of options $x_1, \ldots, x_n$ and asks the user to select their preferred option, i.e., the one with the highest utility. Given a model of how people make such choices, the machine can then infer the underlying function $f$ from the user's chosen item $y$. For example, Stiennon et al. [5] learned a utility function for text summaries by showing users a text with several summaries and asking them to choose the best summary.

There are several models of choice that have been used for learning preferences from human choices. Some recent work on Reinforcement Learning from Human Feedback (RLHF), for example, has used

38th Conference on Neural Information Processing Systems (NeurIPS 2024).

a simple binary choice model [5, 6] $p(y = x_1|x_1, x_2) = \sigma(f(x_1) - f(x_2))$, over choices $x_1$ and $x_2$, though generally, most preference learning approaches have used the Bradley-Terry model [8]

$$p(y = x_i|x_1, \ldots, x_n) = \frac{\exp(\beta f(x_i))}{\sum_{j=1}^n \exp(\beta f(x_j))}.$$

Although these models have proven to be practical, they are not realistic models of human choice behavior. Specifically, both models make choices between two options without taking into account the rank orderings of option attributes; a widely observed property of human decision-making [9–11]. As a result, these models fail to predict a number of apparent biases in human choice behavior. These include contextual choice effects [12, 13], which occur in situations where a decision maker's choice between two options is influenced by adding more options to the choice set [14, 12]. Say, for example, we have two options $A$ and $B$ and a user exhibits a probability of choosing between these. When a third *decoy* option $C$ is introduced which is strictly dominated by $B$, there is a shift in the probability of choices from $A$ to $B$.

Though context effects are not certain to appear in preference queries posed to users, they are known to appear in a wide range of human tasks including risky choice tasks [12], multi-attribute choice tasks [15] and perceptual judgement tasks [11] and in many other species including jays and honeybees [16]. These effects point to a potential gap in the accuracy of the models currently used, during preference learning, to interpret the choices made by users. Moreover, this gap has the potential to lead to incorrect inferences about the latent preference utilities of observed human decision-makers.

The contribution of this paper is threefold. First, we show that we can improve preference learning by leveraging computational rationality theory, a general cognitive-scientific theory which posits that human behavior is rational under cognitive bounds [17, 18]. We learn preferences from human choice behaviors using a state-of-the-art cognitive model that is based on a computational rational analysis of context-dependent choice under uncertainty and is backed by substantial empirical support in the psychology literature [19]. Like all computationally rational models, behavior under this model emerges from the latent utility function and a latent set of cognitive bounds. This provides strong inductive biases when inferring these latent factors from human behavior which – as we will show experimentally – significantly improves learning from preferences. Our second contribution lies in making this cognitive model amenable to preference learning. To this end, we generalize it, and make inference practical by approximating intractable calculations with a surrogate we call the Computationally Rational Choice Surrogate (CRCS) model. Finally, we find experimentally that CRCS can sometimes perform worse than the Linear Context Logit (LCL) model [20]. We hypothesize that human context effects are partially a consequence of cross-feature effects. These are not modeled in CRCS, but can be learnt by LCL. We therefore propose LC-CRCS, which takes advantage of these effects by combining CRCS with LCL.

We report three sets of experiments. In the first, we show that CRCS matches the original model's prediction of human choice behavior. In the second, we compare preference learning with CRCS to preference learning with recently proposed variants of the Bradley-Terry choice model. Using existing human data sets, we show that CRCS outperforms these in choice prediction and utility function inference, but performs worse than LCL on some tasks. We then show that LC-CRCS can additionally outperform LCL in these tasks. In the third set of experiments, we show the applicability of CRCS in three real-world use cases and verify its parameter recovery capability.

## 2  Background

### 2.1  Learning from Preferences

Preference learning methods aim to infer latent utility functions from human choices. Depending on the type of queries presented, there are two main streams of research: (1) learning from pairwise comparisons or (2) learning from ranking, where humans rank a set of $n$ options. Popular methods include Gaussian Process regression that captures the preference relationships of pairwise queries [21, 22]. Other work, such as as [23–25], uses Deep Neural Networks trained on ranked demonstrations to approximate the underlying reward functions. To reduce the computational burden created by the necessity for numerous queries, active learning techniques [26–29] have been proposed for efficient query proposal with maximum information gain. However, these methods typically require consistent preference order within the ranking and do not consider any contextual effects within the query

dataset. Reinforcement Learning approaches include Preference-Based Reinforcement Learning (PBRL) and Reinforcement Learning with Human Feedback (RLHF), where the reward function is inferred from the preference feedback. Work using ranking queries and human feedback can reach or even exceed human-level performance in several RL benchmarks [23, 24].

## 2.2 Modeling contextual choice

To date, preference learning research has yet to make use of plausible models of human decision-making such as [30, 31]. These models are inspired by extensive studies of human behavior and give rise to *contextual choice effects*. Consider a hypothetical choice between two sightseeing trips, one to Paris and the other to London. Both trips come with free coffee. Let's say that $70\%$ of people prefer Paris and $30\%$ London. Now imagine that we add a third option which is identical to the London trip but without free coffee. If, for this three-trip choice problem, we observe that $40\%$ prefer London with free coffee, then we will have observed a contextual choice effect known as a "preference reversal" [32]. The choice frequency for London with coffee is increased by a context that includes a dominated choice. This effect has been observed both in sample averages and, more interestingly, within individuals. It has been taken as evidence that people are irrational [33] and have no stable preferences [34]. Needless to say, both instability and irrationality pose severe challenges to the viability of preference learning.

More recent theories, however, demonstrate that contextual choice effects can be consequences of computationally rational processes that assume stable preferences. These theories explain contextual choice effects by modeling the fact that people compare attributes and/or utilities under uncertainty. These include Bayesian theories [35], rational analyses [19], and neurobiological relative encoding theories [13, 36]. These theories use comparisons between option attributes to compute expected utilities, such that these expectations are sensitive to the reliability of the comparison as an indicator of expected utility.

Other work has proposed variations on the Bradley-Terry model to include these contextual effects, with the same commitment to stable preferences. Bower and Balzano [37] posit that context effects are the result of humans comparing options only on the $k$ most salient features within a context. They propose a Bradley-Terry model where utilities are calculated only on the $k$ most salient features, where saliency is measured by the sample variance of each feature within the current set of options. Tomlinson and Benson [20] do not propose a specific theory of context effects, but rather propose to learn them from data. They introduce the Linear Context Logit (LCL) model, a Bradley-Terry model with a linear utility function, in which context effects are modeled as a context-dependent change in the (globally stable) weights of the utility function. This change is modeled as a linear function of the average attribute values of the set of options presented to a user (the context), and is inferred from human choice data. They further introduce the Decomposed LCL model, in which each feature induces its own context effect – whereas in LCL the features jointly induce a single context effect – and where the final context effect results from a mixture of these individual effects.

In the current paper, we commit to distinguishing behavioral choices, which are observable, from latent preferences, which are not. When we refer to "preferences" we are referring to the latent utility function $f(x)$, and not to the observable choice behavior.

## 3 Modeling computationally rational choice

To learn latent preferences from human choice behavior, we build on a computationally rational model of choice behaviors by Howes et al. [19] which is sensitive to the aforementioned context effects. This model assumes that humans make utility-maximizing choices, but that the option utilities are estimated from noisy observations of the true utilities and noisy comparisons between the option attributes. Here we will first describe the original model in a general form. We then extend it to a general space of utility functions and introduce our Computationally Rational Choice Surrogate (CRCS) model, a model which replaces intractable computations in the original model with learned surrogates to allow tractable inference of the latent utility function. Finally, we introduce the LC-CRCS model, an extension of the CRCS model which is able to learn additional context effects not captured by the CRCS model.

## 3.1 A computationally rational model of choice

Let $x_1, \ldots, x_n \in \mathbb{R}^d$ be a set of $n$ options, each with $d$ attributes. Let $f : \mathbb{R}^d \to \mathbb{R}$ be a latent utility function that maps each option to its associated utility. As a shorthand, we will denote the utilities of a collection of options $\boldsymbol{x} = \langle x_1, \ldots, x_n \rangle$ as $\boldsymbol{u} = \langle u_1, \ldots, u_n \rangle$ where $u_i = f(x_i)$.

The cognitive model introduced by Howes et al. [19] assumes that when making choices, humans do not observe the options $x_1, \ldots, x_n$ nor their utilities directly. Instead, humans are assumed to make utility-maximizing choices based on two sets of noisy observations of the options. The first set are noisy observations $\widetilde{\boldsymbol{u}} = \langle \widetilde{u}_1, \ldots, \widetilde{u}_n \rangle$ of the true utility of each option. These are modeled as samples from a Gaussian centered around the true utilities

$$\forall i \in 1, \ldots, n : \quad \widetilde{u}_i \sim \mathcal{N}(f(x_i), \sigma_{calc}^2)$$

with noise $\sigma_{calc}^2$ which we will call the *calculation noise*. The second set are noisy observations of the ordinal relation between the values of each attribute for each pair of options. Given an attribute $k$ and a pair of options $(x_i, x_j)$, this ordinal relationship is defined by the following observation function:

$$o(x_{i,k}, x_{j,k}) = \begin{cases} \prec & \text{iff } x_{i,k} < x_{j,k} - \tau_k \\ \succ & \text{iff } x_{i,k} > x_{j,k} + \tau_k \\ \equiv & \text{else} \end{cases}$$

with $\tau_k$ an attribute-specific tolerance parameter. Intuitively, a larger $\tau_k$ creates a greater margin within which attribute values will be considered equal. For binary attributes, we set $\tau_k$ to zero. Each noisy ordinal observation $\widetilde{o}(x_{i,k}, x_{j,k})$ is sampled as follows: with probability $1 - \varepsilon$ sample $\widetilde{o}(x_{i,k}, x_{j,k}) = o(x_{i,k}, x_{j,k})$, otherwise sample uniformly at random from $\{\prec, \succ, \equiv\}$. The *probability of ordinal error* $\varepsilon$ is a parameter, and is the sole source of noise within the ordinal observations. We will denote the set of noisy ordinal observations as $\widetilde{\boldsymbol{o}} = \{\widetilde{o}(x_{i,k}, x_{j,k})\}_{k=1\ldots d, i=1\ldots n, j=i+1\ldots n}$.

Given these observations $\widetilde{\boldsymbol{u}}$ and $\widetilde{\boldsymbol{o}}$ for options $x_1, \ldots, x_n$, and the choice model parameters $\theta = (\sigma_{calc}^2, \varepsilon, \tau_1, \ldots, \tau_d)$, the above model implies a posterior distribution over the options' true utilities $p(\boldsymbol{u}|\widetilde{\boldsymbol{u}}, \widetilde{\boldsymbol{o}}, \theta)$ and associated expected values $\mathbb{E}[u_i|\widetilde{\boldsymbol{u}}, \widetilde{\boldsymbol{o}}, \theta]$. As they do not observe true utilities of the options, humans are assumed to choose the option $y$ with the highest expected utility:

$$y = \underset{x_i \in \{x_1, \ldots, x_n\}}{\mathrm{argmax}} \mathbb{E}[u_i|\widetilde{\boldsymbol{u}}, \widetilde{\boldsymbol{o}}, \theta].$$

Preference learning requires that we are able to reason about how various utilities lead to different choice behaviors. Therefore, to make the original cognitive model amenable to preference learning, we replace the fixed utility function $f$ by a space of utility functions $\{f_w\}_{w \in \mathcal{W}}$ parameterized by a utility parameter $w$. We assume that the user being modeled makes choices based on some chosen parameter value $w$, which is known only to them, and which we represent as an additional observed random variable in the model. Necessarily, any calculation of utility therefore depends on $w$. Under these assumptions the user's posterior over utilities, and thus their choice $y$, is:

$$y = \underset{x_i \in \{x_1, \ldots, x_n\}}{\mathrm{argmax}} \mathbb{E}[u_i|\widetilde{\boldsymbol{u}}, \widetilde{\boldsymbol{o}}, w, \theta]. \tag{1}$$

where this expectation is calculated under the posterior

$$p(u|\widetilde{\boldsymbol{u}}, \widetilde{\boldsymbol{o}}, w, \theta) \propto p(\widetilde{\boldsymbol{u}}|\boldsymbol{u}, \theta) \int_{\boldsymbol{x}} p(\boldsymbol{x}, \boldsymbol{u}, \widetilde{\boldsymbol{o}}|w, \theta) d\boldsymbol{x}$$

$$= \prod_{i=1}^{n} p(\widetilde{u}_i|u_i, \theta) \int_{\boldsymbol{x}} p(\widetilde{\boldsymbol{o}}|\boldsymbol{x}, \theta) \prod_{i=1}^{n} p(x_i) p(u_i|x_i, w) d\boldsymbol{x}. \tag{2}$$

## 3.2 Learning from choice behaviors

In our description of the model above, we have taken the point of view of the user making the choices. However, we now return to a preference learning perspective, i.e. that of an outside observer such as an AI system trying to infer the utility function that underlies these choices. We assume that the AI system observes the presented options $x_1, \ldots, x_n$, as well as the option $y$ the user chooses. The goal is then to infer the unknown utility parameter $w$ and choice model parameters $\theta$ from observed

choices $(\boldsymbol{x}, y)$. However, the noisy observations $\widetilde{\boldsymbol{u}}$ and $\widetilde{\boldsymbol{o}}$ on which the user bases their choice are part of their internal perception of the options, and are therefore not observable to an AI system. This means that in evaluating the likelihood of a choice $y$ under the above choice model we must treat the observations as latent. This yields the following choice policy for the user:

$$p(y|\boldsymbol{x}, w, \theta) = \int_{\widetilde{\boldsymbol{u}}} \int_{\widetilde{\boldsymbol{o}}} p(y|\widetilde{\boldsymbol{o}}, \widetilde{\boldsymbol{u}}, w, \theta) p(\widetilde{\boldsymbol{o}}|\boldsymbol{x}, \theta) p(\widetilde{\boldsymbol{u}}|\boldsymbol{x}, w, \theta) d\widetilde{\boldsymbol{o}} d\widetilde{\boldsymbol{u}}. \tag{3}$$

Here, $p(y|\widetilde{\boldsymbol{o}}, \widetilde{\boldsymbol{u}}, w)$ is a point mass on $y$ following equation (1). Given $m$ pairs $(\boldsymbol{x}^{(l)}, y^{(l)})$, a prior $p(w)$ over the space of utility parameters and a prior $p(\theta)$ over the space of choice model parameters, we can use the likelihood in equation (3) to infer a posterior over the parameters $w$ and $\theta$:

$$p(w, \theta | \{(\boldsymbol{x}^{(l)}, y^{(l)})\}_{l=1}^{m}) \propto p(w)p(\theta) \prod_{l=1}^{m} p(y^{(l)}|\boldsymbol{x}^{(l)}, w, \theta).$$

### 3.3 Tractable inference through surrogates

The issue we face in calculating $p(w, \theta|\{(\boldsymbol{x}^{(l)}, y^{(l)})\})$ is that the likelihood $p(y|\boldsymbol{x}, w, \theta)$ is intractable. First, the calculation of the expected values in equation (1) requires the evaluation of an intractable integral over $\boldsymbol{x}$ in equation (2). The expected values can be approximated using a Monte Carlo estimate [19], but many samples are needed to achieve a good approximation. Second, the calculation of the likelihood itself requires the evaluation of an intractable integral over all possible observations in equation (3). As before, one could approximate this integral using a Monte Carlo estimate, but this would again require many samples.

Instead, we propose to train surrogate neural networks to approximate both these quantities. We introduce a first neural network $\widehat{u}(\widetilde{\boldsymbol{u}}, \widetilde{\boldsymbol{o}}, w, \theta)$ trained to predict a vector of the expected values $\mathbb{E}[\boldsymbol{u}|\widetilde{\boldsymbol{u}}, \widetilde{\boldsymbol{o}}, w, \theta]$ from given observations $\widetilde{\boldsymbol{u}}, \widetilde{\boldsymbol{o}}$ and parameters $w$ and $\theta$. Then $\widehat{u}(\cdot)$ is trained by minimizing

$$\mathcal{L}_{\text{util}}(\widehat{u}) = \underset{p(w, \theta, \boldsymbol{u}, \widetilde{\boldsymbol{u}}, \widetilde{\boldsymbol{o}})}{\mathbb{E}} [\|\widehat{u}(\widetilde{\boldsymbol{u}}, \widetilde{\boldsymbol{o}}, w, \theta) - \boldsymbol{u}\|_2]. \tag{4}$$

Samples $(w, \theta, \boldsymbol{u}, \widetilde{\boldsymbol{u}}, \widetilde{\boldsymbol{o}})$ are obtained by (1) sampling $w \sim p(w)$, $\theta \sim p(\theta)$ and $\boldsymbol{x} \sim p(\boldsymbol{x})$ from their respective priors, (2) calculating $u_i = f_w(x_i)$ for each option, and (3) sampling the observations $\widetilde{u}_i \sim p(\widetilde{u}_i|u_i, \theta)$ and $\widetilde{\boldsymbol{o}} \sim p(\widetilde{\boldsymbol{o}}|\boldsymbol{x}, \theta)$. Note that the minimum of $\mathcal{L}_{\text{util}}(\widehat{u})$ is exactly the function that assigns to each tuple $(\widetilde{\boldsymbol{u}}, \widetilde{\boldsymbol{o}}, w, \theta)$ the vector of expectations $\mathbb{E}[\boldsymbol{u}|\widetilde{\boldsymbol{u}}, \widetilde{\boldsymbol{o}}, w, \theta]$.

Next, we train a second neural network $\widehat{q}(y|x, w, \theta)$, which we will refer to as our *CRCS model*, to approximate the user's policy $p(y|\boldsymbol{x}, w, \theta)$ over choice behaviors. By using the fact that $\widehat{u}(\widetilde{\boldsymbol{u}}, \widetilde{\boldsymbol{o}}, w, \theta) \approx \mathbb{E}[\boldsymbol{u}|\widetilde{\boldsymbol{u}}, \widetilde{\boldsymbol{o}}, w, \theta]$ we minimize the cross-entropy loss between $\widehat{q}$ and choices based on utilities predicted by $\widehat{u}$. The loss function is thus:

$$\mathcal{L}_{\text{pol}}(\widehat{q}) = \underset{p(w, \theta, \boldsymbol{x}, \widetilde{\boldsymbol{u}}, \widetilde{\boldsymbol{o}})}{\mathbb{E}} \left[ -\ln \widehat{q} \left( \underset{\{x_1, \dots, x_n\}}{\operatorname{argmax}} \widehat{u}(\widetilde{\boldsymbol{u}}, \widetilde{\boldsymbol{o}}, w, \theta) \middle| \boldsymbol{x}, w, \theta \right) \right].$$

Samples $(w, \theta, \boldsymbol{u}, \widetilde{\boldsymbol{u}}, \widetilde{\boldsymbol{o}})$ are obtained as above.

### 3.4 Modeling cross-feature influence in CRCS

Although our proposed model can predict a range of context effects, it does not yet capture all. Although CRCS can model how each individual feature influences the expected utility of the options, it cannot model how features can impact each other. This is something that LCL does do: its utility weight updating mechanism changes the weight of each feature based on the mean value of all other features. Thus, features can influence how other features are valued. In the most general sense, LCL's fundamental mechanism corresponds to a function $g(w, \boldsymbol{x})$ which maps the utility weights $w$ and the set of options $\boldsymbol{x}$ (which make up the context) to a new set of weights $w'$. We therefore propose to integrate this same mechanism into CRCS, resulting in a new model $\widehat{q}(y|\boldsymbol{x}, g(w, \boldsymbol{x}), \theta)$. As $\widehat{q}$ is differentiable, we can infer $g$ from data using gradient descent. In the experiments that follow, we will use this approach in settings where $f_w$ is a linear function. Thus, like LCL, we will define $g$ as a linear function of $x_C$: the mean attribute values of the options $\boldsymbol{x}$. We will refer to the resulting model $\widehat{q}(y|\boldsymbol{x}, w + (Ax_C)^T, \theta)$ as LC-CRCS.

Table 1: Choice model NLLs on human choice data sets. Bolded digits indicate a significant (p < 0.01) improvement over baselines (BT, BB, LCL).

| Dataset | Bradley-Terry | Bower & Balzano | LCL | CRCS (ours) | LC-CRCS (ours) |
|---|---|---|---|---|---|
| Hotels | 573 | 573 | 553 | **536** | **536** |
| District-Smart | 3432 | 3432 | 3305 | 3371 | **3276** |
| Car-Alt | 7414 | 7416 | 7290 | 7322 | 7345 |
| Dumbalska | 103669 | 103711 | 100683 | 100450 | **99147** |

## 4 Experiments

We first validate the proposed CRCS model by comparing its results with the original computationally rational choice model by Howes et al. [19]. Then we compare the proposed CRCS model and our LC-CRCS variant with three baselines on human choice data, and finally study the performance of the model on three case studies: car crash structure design, water drainage network design, and retrosynthesis planning.[1]

We evaluate our proposed CRCS model on four datasets of human choices. These datasets are large sets of choices $(x^{(l)}, y^{(l)})$ collected from human participants. The District-Smart dataset [38] contains pairwise preferences over voting districts, where participants were asked to choose the district they felt was most compact. The features extracted for each district are six geometric measures identified by the original authors as good measures of compactness. The Car-Alt dataset [39] contains choices between six hypothetical alternative fuel cars. Each car has 21 features, including size, range, operating cost, etc. We also use a dataset collected in [40], which we will call the Hotels dataset, where in a user study participants were asked which of three hotels they preferred. The hotels were collected from a booking site and had as features the price per night and average review rating. For each participant, a choice was collected on one of six sets of options constructed to target three known context effects: attraction, compromise and similarity. Lastly, we use the data collected by Dumbalska et al. [36] on a property task, which we will refer to as Dumbalska. Here, participants ranked three properties in order of best to worst value. For our purposes, we will treat the top-ranked item as the choice. Value was defined as the given rental cost minus the value participants thought the house was worth (which had been elicited in an earlier stage). For each participant, responses were collected on a large collection of choices, specifically engineered to span the entire range of potential context effects. Thus, unlike the other datasets, we have multiple recorded choices per participant. This allows us to make inferences per individual, rather than at the population level, and evaluate how well our choice models fit the preferences and context effects exhibited by individuals.

### 4.1 Validation of the CRCS model: risky choice tasks with preference reversals

In this experiment, we validate our CRCS model against the original implementation of Howes et al. [19] on a risky choice task. In this task, a user is presented with a set of three options, each of which is a pair $(p_i, v_i)$ consisting of a probability $p_i$ and a payoff $v_i$. Upon selecting option $i$, the user receives payoff $v_i$ with probability $p_i$, meaning that each option has expected payoff $f(p_i, v_i) = p_i \cdot v_i$.

Comparing expected option values predicted by $\widehat{u}$ with the Monte Carlo estimates used in [19], we find that on sets of three options, both generally agreed on the relative magnitude of the utilities, and agreed on the ranking of the utilities in $92.277\% \pm 0.165\%$ (Agresti–Coull) of cases. Next, we verified $\widehat{q}$'s ability to predict contextual preference reversals. This was tested on Range-Frequency decoy conditions [19] where two "Pareto-optimal" options with equal utility are presented along with a decoy option with slightly lower utility which is dominated by one of the other two options. Preference reversals – specifically, increased likelihood of choosing the Pareto-optimal option that dominates the decoy – have been observed in humans and are predicted by the original model. Figure 3 in the appendix shows that $\widehat{q}$ reproduces the range of reversal rates of the original model.

---

[1]Implementation available at `https://github.com/AaltoPML/Preference-Learning-with-a-human-like-model-of-choice`.

Table 2: Consistency of inferred utility function with separately collected rankings on District-Smart. Bolded digits indicate a significant improvement over baselines (BT, BB, LCL).

| Dataset | Bradley-Terry | Bower & Balzano | LCL | CRCS (ours) | LC-CRCS (ours) |
|---|---|---|---|---|---|
| District-Smart | 0.162 | 0.217 | 0.286 | **0.622** | **0.525** |

## 4.2 Evaluation on static human choice data

In this set of experiments, we evaluate each models' ability to generalize to unseen data. We compare our proposed CRCS model and the LC-CRCS variant against three baselines: vanilla Bradley-Terry, the variant proposed by[37] (referred to as Bower & Balzano) and LCL [20]. On four different datasets, we infer the parameters for each model on a training set of observed choices $\{(\boldsymbol{x}^{(l)}, y^{(l)})\}_{l=1}^{m}$ and calculate the negative log-likelihood (NLL) of a held-out test set under the inferred parameters. Inference was done using gradient descent on the NLL of the training set. We performed cross-validation, and report the sum of the test sets' NLLs across the folds. For Hotels, Car-Alt and District-Smart we split the choice data across 50, 20 and 10 folds respectively. By evaluating each choice model on each test fold, we obtained paired observations (one per condition) for each test fold, allowing us to perform a Wilcoxon rank test across the folds to test significance. On Dumbalska, we look at how well the choice models can fit to individuals, and thus perform cross-validation for each participant individually. We then treated the sum of the NLLs of the test sets per participants as individual measures, and tested significance using a Wilcoxon test across the participants. Following prior work, we used a linear utility function in all choice models on all datasets.

Table 1 shows the total NLL achieved by each model on each dataset. We observe that our proposed LC-CRCS model achieves the highest NLL on Hotels, District-Smart and Dumbalska. This difference is significant ($p < 0.01$) in all three cases. On Car-Alt, we see that LCL performs better than all other models, with the difference being significant ($p < 0.01$) for all except the CRCS model ($p > 0.2$). We theorize that the poor performance of the CRCS model on Car-Alt is due to insufficient option data to train $\widehat{u}$ on (see Appendix A.1), leading to poor estimates of expected utility and therefore poor choice predictions.

### 4.2.1 Evaluating the inferred utility function

As part of the District-Smart human subject study, Kaufman et al. [38] collected rankings on six sets of districts from small groups of participants. Ranking such large sets is quite difficult, and we should expect these rankings to be quite noisy. However, like the binary choices that were collected, these rankings are indicative of people's true preferences, and thus should be consistent with any ranking of the same districts implied by the utility function we infer from the binary choices. To test this, we use our choice models to infer utility parameters on the entire set of binary choices. For each of the six sets, we then measure – using Kendall's $\tau$ [41] – how consistent the ranking implied by the inferred utility parameters is with the ranking collected in the study. We report the average consistency across all six sets. Because the log-likelihood of CRCS and LC-CRCS is not convex, we repeat this procedure 25 times, starting from different points, to control for the effect local optima may have on the inferences. We test significance using a Wilcoxon test across the six sets of rankings.

Unlike the previous experiment, during inference we regularized the choice model parameters of LCL, CRCS and LC-CRCS. This was essential to infer utility parameters that were consistent with the collected rankings. For LCL we used the L1 matrix norm of the weight adaptation mechanism's parameter matrix as a regularization term. The L1 norm enforces sparsity and thus encourages LCL to fit only to the most significant context effects [20]. For CRCS and LC-CRCS we used the probability of the choice model parameters under a chosen prior as the regularization term. Using our understanding of the model this allowed us to encode specific prior knowledge into the regularization. More details can be found in Appendix A.3. From the results in Table 2 we observe that both CRCS and LC-CRCS infer utility parameters that are significantly ($p < 0.001$) more consistent with the collected rankings than the baselines. LC-CRCS performs slightly worse than CRCS, though the difference was not yet significant.

### 4.3 Elicitation on human choice data

We now evaluate how well the choice models perform in a preference learning setting, where we actively select the queries we put to a user. Whereas in the previous set of experiments we evaluated how well the choice models perform on large amounts of data, here we are interested in how well they perform when minimal data is available. We use the wealth of data available per participant (between 530 and 1060 responses) in the Dumbalska data set to run a user experiment in silico. For each choice model, we use active learning to infer utility function and choice model parameters for each participant individually. At each time step of the experiment, we select from the set of queries recorded for the participant the most informative query using the expected information gain. The participant's response to this query is then revealed, and the posterior over the utility and choice model parameters is updated. We use a particle filter to maintain the posterior beliefs. This elicitation process is performed for 25 time steps on 75 participants. To evaluate the inferences made by the choice models, we calculate the expected likelihood of the remaining queries where the choice has not been revealed yet. As the training data is actively selected for each choice model independently, the data on which they are evaluated – the remaining queries – will differ, meaning that we cannot use a paired test as we have done so far. Instead, we test significance using an independent t-test.

Figure 1a shows the mean expected utility calculated over the participants as a function of time for each choice model. We observe that the two variants of our CRCS model make significantly (p < 0.01) better predictions of the participants' choices at all time steps (except 0). Where in the previous experiment (Table 1) we saw that LCL was close in predictive power to our proposed models, we see that in this low data setting the difference is much more pronounced. This is because a number of the context effects observed in the Dumbalska data set are built into the CRCS model. While LCL has shown it can learn some of these effects, it needs much more data to do so. Interestingly, we also observe that the LC-CRCS model, which can learn some new context effects on top of the ones built into $\widehat{q}$, shows significant (p < 0.05) improvement on the CRCS model itself, even when very little data is available. This shows that it provides us with the best of both worlds, showing quick adaptability in low data settings and good performance when more data is available.

### 4.4 Simulated case studies

We now test the feasibility of using our model to learn a utility function from simulated choice behaviors in real-world tasks, and to use the inferred utility function to help a designer solve a task by recommending design solutions to them. We consider a preference learning setting where we learn utility and choice model parameters by iteratively eliciting a simulated designer's preferences over sets of candidate designs, chosen to maximize the expected information gain. Using $\widehat{q}$, we infer a posterior over the unknown parameters from the observed choices. To simulate a variety of users, we run this experiment in silico, using $\widehat{q}$ with utility parameters sampled from a non-informative prior and choice model parameters sampled from a prior designed to capture a wide range of behaviors exhibited by the CRCS model. At each time step we measure two things: our ability to recover the unknown parameters from the observed choices, and the utility of the design recommendations we make. The inference error is measured by the distance to the true parameters under our current posterior beliefs. The second is measured using the recommendation regret: the difference in utility between the designer's optimal design and the recommended design, which is chosen to be the design with the highest expected utility under the posterior.

#### 4.4.1 Case study 1: learning from preferences in structural design

The first use case involves the design of the frontal crash structure of a car to optimize three separate objectives $g_1(\boldsymbol{t}), g_2(\boldsymbol{t}), g_3(\boldsymbol{t})$ [42]. The design is parameterized by five parameters $\boldsymbol{t} = \langle t_1, \ldots t_5 \rangle$ which determine the thicknesses of various metal elements. We define the utility function as the Chebyshev scalarization of the original objectives: $f_w(\boldsymbol{t}) = \max_{i \in \{1,2,3\}} w_i |g_i(\boldsymbol{t}) - z_i^*|$ where $z_i^*$ denotes the ideal value of $g_i()$ and the weights $w$ sum to one. Different choices of the utility weights $w$ correspond to different trade-offs between the objectives, and therefore to different solutions on the Pareto frontier. Figure 1b shows the average recommendation regret as a function of the number of queries across 300 runs of this experiment. We observe that the recommendation regret reduces quickly, yielding good recommendations after as few as 10 queries. We attribute this to the utility inference error, shown in Figure 6a in Appendix B.1, which reduces equally quickly.

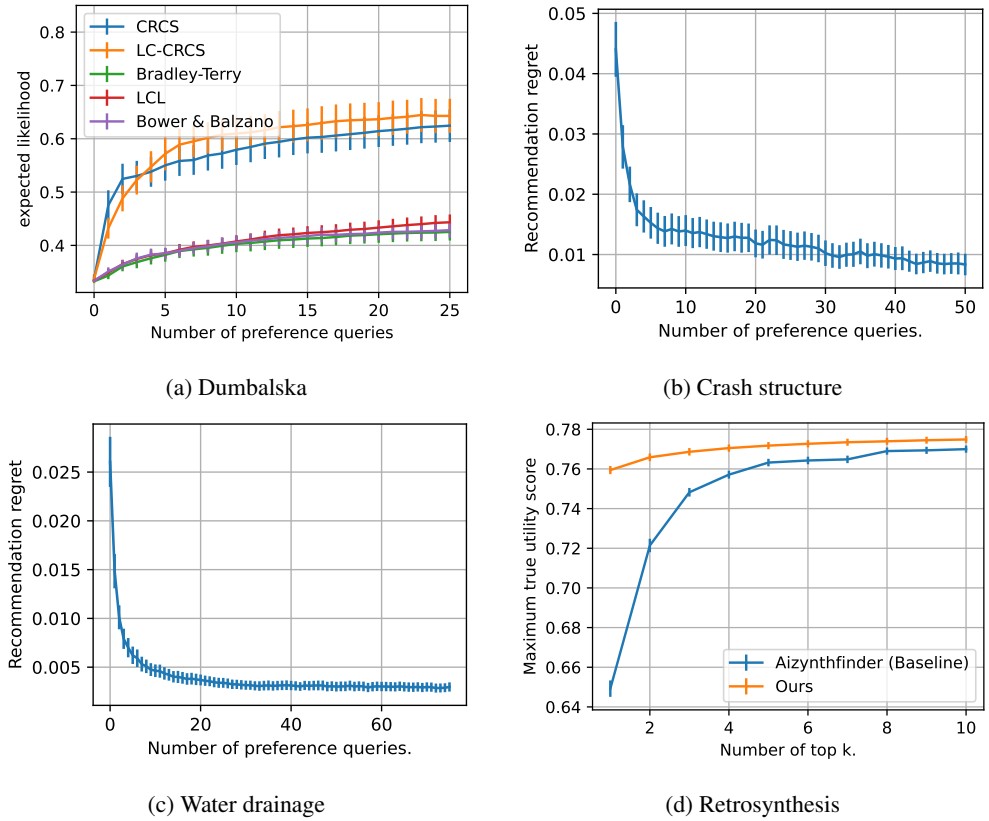

(a) Dumbalska

(b) Crash structure

(c) Water drainage

(d) Retrosynthesis

Figure 1: (a) Mean expected likelihood of unseen choice data as a function of the number of queries observed for various choice models on the Dumbalska elicitation task. (b-c) Mean recommendation regret as a function of the number of queries observed for the crash structure design and water drainage network design respectively. (d) Maximum utility within the top $k$ of routes ranked by inferred utility as a function of $k$. All plots show the mean $\pm$ twice the standard error around the mean.

#### 4.4.2 Case study 2: learning from preferences in water drainage network design

Our second use case is a water drainage network design problem [43]. This use case is another multi-objective problem involving six objectives $g_1(\boldsymbol{z}), \ldots, g_6(\boldsymbol{z})$. Here, we scalarize the problem using a weighted sum $f_w(\boldsymbol{z}) = \sum_{i=1}^{6} w_i g_i(\boldsymbol{z})$ where the weights $w$ sum to one. As before, different choices of $w$ correspond to different solutions on the Pareto frontier. We ran 300 runs of this experiment. Here too we see that recommendation regret (Figure 1c) and utility inference error (Figure 6a in the appendix) drop quickly as we put more queries to the designer, with the largest reduction within the first 10 queries. We achieve high-quality recommendations after less than 10 queries.

#### 4.4.3 Case study 3: improving retrosynthesis planning with preference learning

Retrosynthesis planning, the problem of finding feasible reaction pathways to synthesize target molecules, is a central task of synthetic chemistry. Significant progress has been made in solving it through end-to-end automatic synthesis planning [44–46]. Existing work has focused on expanding the search space of feasible reaction plans. Each route may satisfy an additional subset of properties, and different individuals or organizations may have different preferences over the properties. Chemists' preferences over these plans are often highly complex, representing trade-offs between multiple objectives informed by personal experience and company policy. However, learning their preferences in a way that can then inform AI-driven synthesis planning has not yet been done. We designed a chemist-in-the-loop retrosynthesis planning framework to generate routes with an inferred user model.

To build a personalized retrosynthesis planner, we modified one of the state-of-the-art automatic retrosynthesis platforms, Aizynthfinder [45], built on Monte Carlo Tree Search (MCTS) with a fixed utility function. Details can be found in Appendix B.2.1. First, we proposed a new utility function as the weighted combination of five feature properties $g_1(\boldsymbol{r}), \ldots, g_5(\boldsymbol{r})$, that correspond to *reactants cost, intermediates stability, reaction feasibility, total reaction success rate, poor reaction success rate*, and a route score computed by a data-driven scoring model $g_5$. Given the input routes, this model predicts the distance between the current route and the (latent) optimal route. We trained this model on 47,055 synthetic routes extracted from the Journal of Medicinal Chemistry.

We report the inference error during preference learning in Appendix B.2.2. We integrated the inferred utility weights into our planning system and assessed the consistency of the generated route with the ground truth user utility preferences. We used inferred weights to synthesize 100 target molecules for each weight. In order to measure the recommendation quality, we evaluated the top-ranked routes from both Aizynthfinder and our model under the true utilities. Specifically, we measured the maximum true utility score among the list of top $k$ recommendations. This is to show how far down from the recommended options list the user needs to go to find their optimal choice. Figure 7c shows that within the top $k$ options, the reaction pathways recommended from our model reaches higher maximum utility score compared to the ones generated by Aizynthfinder. As a significance test, we use the Wilcoxon rank test across every molecule and every user utility with $p < (1.61 \times 10^{-53})$ for all $k$.

## 5 Conclusion

In this paper we have proposed a tractable surrogate model of choice, called CRCS, inspired by theories of human decision-making. This model was shown to be a better basis for preference learning than some, but not all, existing models. In response, we modified the model so that it could make cross-feature observations of feature values extending the opportunity for contextual decision-making. We verified against human data from a range of tasks that the new model, called LC-CRCS, outperforms the tested models both in terms of its ability to predict choices and in its inferences of the utility function that underlies the observed choices. Moreover, we find that it corresponds well to previously reported experimental data demonstrating human susceptibility to contextual choice effects. Feasibility of using the new model for preference learning and its ability to recover parameters was also demonstrated in three case studies. Together, the results demonstrate the viability of CRCS and LC-CRCS in high performance preference learning systems.

**Limitations and future work**    We identify two primary limitations. First, training CRCS requires sufficiently many choice sets, or a sufficient well-specified task so that new sets can be generated. As we saw with Car-Alt, when insufficient choice sets are available for training, performance can suffer. Second, CRCS and CRCS-LC only work on choice sets of fixed size. Extending these surrogates to variable size choice sets is a promising direction for future work. Another promising direction for future work is the application of the current choice model to large language model (LLM) fine-tuning. Currently, given some featurization of LLM responses to a prompt, CRCS could be directly applied. However, this would ignore the reading and interpreting of these responses that human evaluators have to do. As such, we see an extension of the current choice model that integrates these cognitive processes, based on the same computational rationality theory, as potentially transformational future work.

**Societal impact**    This paper presents work motivated by the goal to advance the field of Machine Learning. The potential societal impact is in line with the broad body of prior work on learning from preferences and modeling humans, none of which we feel must be specifically highlighted here.

**Acknowledgements**    This work was supported by the Research Council of Finland (flagship programme: Finnish Center for Artificial Intelligence, FCAI; grants 345604, 341763 and 359207), and the UKRI Turing AI World-Leading Researcher Fellowship, EP/W002973/1. Computational resources were provided by the Aalto Science-IT Project.

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

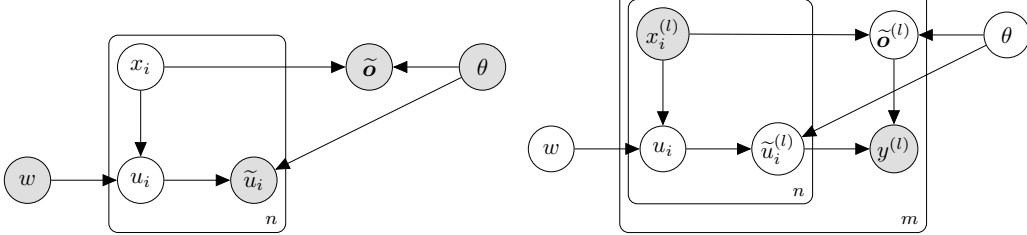

(a) Our choice model, originally introduced in [19], posits that humans make utility-maximizing choices (for some utility function parameters $w$ and choice model parameters $\theta$) based only on observations $(\widetilde{\boldsymbol{u}}, \widetilde{\boldsymbol{o}})$. The options $x_1, \ldots, x_n$ and their true utilities $u_1, \ldots, u_n$ are not observed.

(b) An outside observer observes a set of choices $y^{(l)}$ made over associated options $x_1^{(l)}, \ldots, x_n^{(l)}$. From this data set, the objective is to infer the parameters $w$ and $\theta$. The noisy observations $(\widetilde{\boldsymbol{u}}^{(l)}, \widetilde{\boldsymbol{o}}^{(l)})$ that are central to each of the user's choices are internal to the user and are therefore unobserved.

Figure 2: Graphical models of **(a)** our cognitive choice model and **(b)** the corresponding preference learning problem.

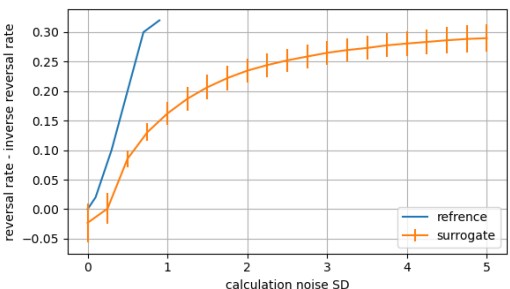

Figure 3: Reversal rate minus inverse reversal rate as a function of $\sigma_{calc}^2$ on Range-Frequency conditions for $\widehat{q}$ ("surrogate") and for the original implementation of Howes et al. [19] ("surrogate"). For $\widehat{q}$, we show the mean $\pm$ std. dev. for 10 models trained with different seeds. The "reversal rate" is measured by calculating the rate at which the Pareto-optimal decoy-dominating option is chosen. To control for random variation we subtract from this the "inverse reversal rate", the rate at which the other Pareto-optimal option is chosen. For non-zero values of $\sigma_{calc}^2$, we see that though $\widehat{q}$ is less sensitive to $\sigma_{calc}^2$, it reproduces the range of reversal rates of the original model.

## A Human data experiments

### A.1 Priors

This section provides details on how the CRCS model was trained for the choice tasks corresponding to the Hotels, District-Smart, Car-Alt and Dumbalska datasets. In order to train our CRCS model on a new choice task, we need to define three priors: a prior over sets of options $p(x)$, a prior over utility function weights $p(w)$, and a prior over choice model parameters $p(\theta)$.

The prior over sets of options $p(x)$ is by far the most important prior for successfully training the CRCS model. It is clearly important that this prior matches the distribution of choice sets we expect to see for the choice task we target. However, it is even more important to ensure that that prior has proper support across the entire space of option sets. From equation 2 we see that in order to predict the true utilities of the options $x$, $\widehat{u}$ essentially has to infer the option set $x$ (which it does not observe) from the observations $\widetilde{u}$ and $\widetilde{o}$. It can only learn to do this well if during training we can expect it to encounter all $x$ that could have resulted in $\widetilde{u}$ and $\widetilde{o}$.

The priors $p(x)$ for these tasks were defined as follows:

- For **Hotels** we had access to the set of 200 hotels the original authors had used to build their study. Thus, we generated option triplets from the prior by uniformly sampling (without replacement) three hotels from this set.

Table 3: Mean $\pm$ std. dev. around the mean for averaged NLL per choice pair on human choice data sets. This table shows the same results as Table 1 but reports averaged as opposed to summed NLLs.

|  | Hotels | District-Smart | Car-Alt | Dumbalska |
|---|---|---|---|---|
| Bradley-Terry | $0.944 \pm 0.109$ | $0.638 \pm 0.015$ | $1.593 \pm 0.022$ | $0.629 \pm 0.259$ |
| Bower & Balzano | $0.944 \pm 0.109$ | $0.637 \pm 0.015$ | $1.593 \pm 0.022$ | $0.629 \pm 0.259$ |
| LCL | $0.910 \pm 0.147$ | $0.614 \pm 0.021$ | $1.563 \pm 0.041$ | $0.613 \pm 0.266$ |
| CRCS (ours) | $0.882 \pm 0.104$ | $0.627 \pm 0.016$ | $1.573 \pm 0.023$ | $0.612 \pm 0.266$ |
| LC-CRCS (ours) | $0.882 \pm 0.115$ | $0.610 \pm 0.020$ | $1.591 \pm 0.029$ | $0.605 \pm 0.273$ |

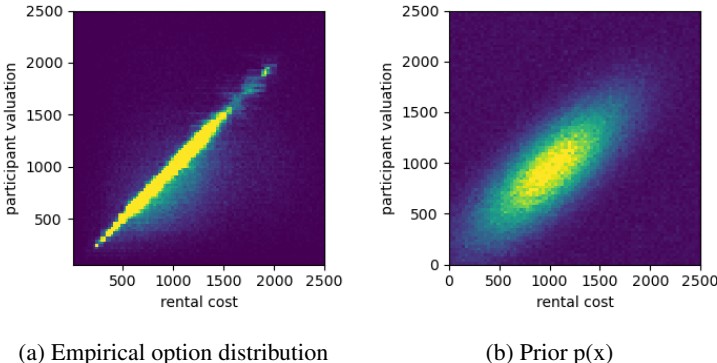

(a) Empirical option distribution        (b) Prior p(x)

Figure 4: (a) The distribution of individual options in the choice data on Dumbalska. (b) The prior $p(x_i)$ over individual options we use to generate new option sets.

- For **District-Smart** we had access to 21778 electoral districts collected by the original authors. We generated pairs of options by sampling them uniformly from this set without replacement.

- Options in the **Dumbalska** task have two features, a property's rental cost and the participant's valuation of it, both of which were bounded between 0 and 2500. The human choice data suggested that both are strongly correlated. We created a prior over individual choices that reproduces this correlation by using a multivariate normal distribution, mixed with a uniform distribution over the entire option space, to ensure sufficient support even on less frequently encountered options. Figure 4 shows the empirical option distribution within the choice data, and our engineered prior over individual options. To generate option sets, we then sampled three options from this prior.

- **Car-Alt** considers options sets consisting of hypothetical cars. Although we know that the options were created from a set of 120 cars, the features of each option are determined both by the Car that corresponds to that option and the participant who makes the choice. For example, the cost of each car is expressed as a multiple of the participant's log income. The inclusion of participant-dependent features creates correlation between the options. Unfortunately, we did not have enough information to engineer a new prior $p(x)$ that would faithfully reproduce this correlation, and would faithfully match the empirical distributions over sets of options encountered in the original user study. Thus, we were forced to train the CRCS model on the limited number of choice sets that appear in the human choice data.

As all tasks used linear utilities, and as the CRCS model is invariant to scaling of the utility function, we define $p(w)$ in all cases as a uniform distribution over all vectors of length 1.

The prior over the choice model parameters was set based on choice model parameters reported for risky choice in [19] after accounting for scale differences of utilities and feature values for each task. They are listed in table 4[2]. For LC-CRCS, we additionally placed independent standard normal priors on all entries of $A$.

---

[2] $\mathcal{N}_a^b$ denotes a normal distribution truncated to the range $(a, b)$. Note that in order to be consistent with other baselines, within the CRCS model utilities are calculated on z-normalized features, while ordinal observations are calculated on non-normalized features.

Table 4: CRCS model parameter priors for various choice tasks.

| | $p(\sigma_{calc})$ | $p(\varepsilon)$ | Attribute $k$ | $p(\tau_k)$ |
|---|---|---|---|---|
| Hotels | $U(0,5)$ | $Beta(1,3)$ | Price per night:
Review rating: | $U(0,100)$
$U(0,1)$ |
| District-Smart | $\mathcal{N}_0^\infty(0.0, 2.0)$ | $Beta(1,3)$ | hull
bbox:
reock:
polsby:
sym_x:
sym_y: | $\mathcal{N}_0^\infty(0.06, 0.12)$
$\mathcal{N}_0^\infty(0.08, 0.16)$
$\mathcal{N}_0^\infty(0.05, 0.1)$
$\mathcal{N}_0^\infty(0.08, 0.16)$
$\mathcal{N}_0^\infty(0.14, 0.28)$
$\mathcal{N}_0^\infty(0.1, 0.2)$ |
| Car-Alt | $\mathcal{N}_0^\infty(0.0, 2.0)$ | $Beta(1,3)$ | Price divided by ln(income):
Range:
Acceleration:
Top speed:
Pollution:
Luggage space:
Operating cost:
Station availability: | $\mathcal{N}_0^\infty(0.7, 1.4)$
$\mathcal{N}_0^\infty(45, 90)$
$\mathcal{N}_0^\infty(1, 2)$
$\mathcal{N}_0^\infty(8.5, 17.0)$
$\mathcal{N}_0^\infty(0.15, 0.3)$
$\mathcal{N}_0^\infty(0.07, 0.15)$
$\mathcal{N}_0^\infty(1.8, 3.0)$
$\mathcal{N}_0^\infty(0.2, 0.4)$ |
| Dumbalska | $\mathcal{N}_0^\infty(0.0, 2.0)$ | $Beta(1,3)$ | Rental cost
Participant's valuation | $\mathcal{N}_0^\infty(200, 400)$
$\mathcal{N}_0^\infty(180, 360)$ |

Table 5: Mean ± std. dev. of averaged NLLs on a randomly selected held-out test set over 20 independent parameter inference runs.

| | Hotels | District-Smart | Car-Alt | Dumbalska |
|---|---|---|---|---|
| Bradley-Terry | $0.890 \pm 8{\times}10^{-6}$ | $0.632 \pm 1{\times}10^{-5}$ | $1.575 \pm 3{\times}10^{-5}$ | $0.567 \pm 3{\times}10^{-6}$ |
| Bower & Balzano | $0.890 \pm 2{\times}10^{-5}$ | $0.631 \pm 4{\times}10^{-6}$ | $1.575 \pm 5{\times}10^{-5}$ | $0.567 \pm 2{\times}10^{-6}$ |
| LCL | $0.831 \pm 2{\times}10^{-5}$ | $0.616 \pm 4{\times}10^{-6}$ | $1.560 \pm 5{\times}10^{-5}$ | $0.562 \pm 2{\times}10^{-6}$ |
| CRCS (ours) | $0.855 \pm 0.014$ | $0.628 \pm 0.009$ | $1.620 \pm 0.032$ | $0.568 \pm 0.007$ |
| LC-CRCS (ours) | $0.902 \pm 0.063$ | $0.616 \pm 0.006$ | $1.590 \pm 0.013$ | $0.572 \pm 0.007$ |

## A.2 Additional details on static data experiments

The experiments on static data were run using a cross-validation strategy. For each fold, we inferred utility parameters and choice model parameters jointly for each choice model by performing vanilla gradient descent on the train set log-likelihood. For the CRCS and LC-CRCS model the starting points were sampled from the priors defined in section A.1. For the Bradley-Terry variants the utility parameters were sampled independently from a standard Gaussian, and for LCL the learnable parameter matrix $A$ was populated using sampled from $\mathcal{N}(0, 0.1)$.

CRCS and LC-CRCS have non-convex likelihood functions, meaning that gradient descent is liable to get stuck in local optima. To mitigate this, we performed inference multiple times (up to 50), starting from multiple starting points, and chose the parameters that achieved the best log-likelihood on a held-out part of the training data. Table 5 shows the variance in average NLL achieved by individual inference runs on the various choice problems. We can see that there is significantly higher standard deviation when doing inference with CRCS and LC-CRCS, pointing to the existence of local minima, and confirming the necessity of repeated inference to address this.

## A.3 Additional details on rank consistency experiments

For District-Smart, we used gradient descent to infer the utility and choice model parameters for each choice model on the entire set of binary choices. For LCL, CRCS, and LC-CRCS we noticed that the inferred utility functions were highly inconsistent with the rankings that had been collected in the same user study. As explained in the main paper, we used regularization to address this. We will go into some more detail on why we think regularization was needed and how we tuned the regularizers

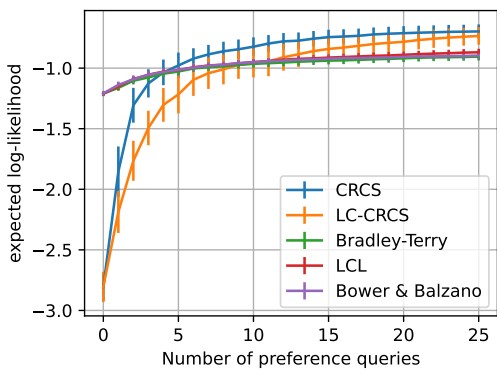

Figure 5: Mean Expected log-likelihood over unseen queries as a function of the number of queries seen. Error bars respond to twice the standard error around the mean.

we used. Table 6 below shows the consistency of the ranking implied by the inferred weights for each of these three choice models with the collected rankings. We can see immediately that compared

Table 6: Consistency between collected rankings and rankings implied by inferred weights with and without regularization of choice model parameter for LCL, CRCS and LC-CRCS.

|  | LCL | CRCS (ours) | LC-CRCS (ours) |
|---|---|---|---|
| Without regularization | -0.36 | -0.17 | -0.04 |
| With regularization | 0.287 | 0.622 | 0.525 |

to Bradley-Terry and the Bower & Balzano models, the consistency is quite poor, especially for LCL. We hypothesize that this is caused by heterogeneity in the task utilities different participants used in making their choices. The intention of Kaufman et al. [38] was to capture humans' intuitive understanding of what it means for an electoral district to be compact. Therefore, in the user study, people were encouraged to make choices "according to your own best judgement" [38]. As it is likely that the various participants in the study had slightly different intuitions about compactness, we can therefore expect that the recorded choices have been made with slightly different utility weights $w$. To fit a choice model using a single utility to these choices could thus prove problematic. For future work, it would be interesting to consider a hierarchical approach, where we would model the fact that any choice has been made according to an unobserved utility function drawn from some unknown distribution.

For the current work, we resorted to using regularization to ensure the choice models did not overfit to the noise in the utility function. These regularization strategies were tuned using one of the six rankings collected, while for evaluating the consistency of the inferred weights with the collected rankings we used all six. For LCL, we regularized the weight update mechanism $w + Ax_C$ by using the norm of $A$, multiplied by 75 to get the desired regularization strength, as a regularization term. For the CRCS model, to ensure that we make sensible inferences, we had to ensure that the noise in the utility function would be explained by the right source of noise in the model. The prior we placed on the choice model parameters was designed to do just this. We used placed a $Beta(1, 1000)$ to ensure that $\varepsilon$, which determines the level of noise on the attribute comparisons, would stay close to 0. Attribute comparisons are the primary source of context effects and are necessary to fit to any such effects in the data. Additionally, we note that in the experiment conducted in [19], $\varepsilon$ was fixed to 0. We then placed a $\mathcal{N}(25.0, 0.1)$ prior on $\sigma_{calc}$, the noise on the utility observations, to help explain the heterogeneity of utility functions itself. We also placed a weak prior on $\tau_1, \ldots, \tau_d$, namely the prior we also used when training the CRCS model. The same priors were used for LC-CRCS.

## A.4 Additional details on elicitation experiment

The elicitation experiment on the Dumbalska dataset was performed for each participant in the original experiment individually. We excluded participants according to the same rule the original paper had used. For each run, we would select a participant from the dataset and treat the queries to which responses had been collected for this participant as the queries we could put to the user. In each time step, we used the posterior at that point to estimate the expected information gain of each query that had not been used yet, and selected the query with the highest information gain. The recorded response to this query would then be revealed, and the posterior would be updated with this new observation using the choice model. To maintain the posterior, we used a particle filter containing up to five million particles representing combinations of utility and choice model parameters. The particle filter was not refreshed during the experiment. At each time step we measured the expected likelihood, where the expectation was taken with regard to the posterior and a uniform distribution over the remaining set of recorded queries (those which had not yet been selected as part of the elicitation process). For completeness, we also measured the expected log-likelihood and the entropy in the marginal utility parameter posterior. Those are shown in Figure 5.

## A.5 Additional information on the datasets used

We list below here the sources for the data we use in the human data experiments for Dumbalska [36], Car-Alt [39], Hotels [40] and District-Smart [38]. We obtained the human choice data for District-Smart and Car-Alt from the excellent collection of choice data collected by Tomlinson and Benson [20] for their implementation.

| Dataset | Location | Filename | Description | License |
|---|---|---|---|---|
| Dumbalska | OSF | decoy_233_participants.mat | human choice Data | CC-By 4.0 Attr. |
| Car-Alt | GDrive | car-alt.zip | human choice data | - |
| Hotels | OSF | data.xls | human choice data and list of hotels used to train CRCS model. | - |
| District-Smart | GDrive | district-smart.pickle | human choice data | MIT |
| | Github | training_data.RData | human rankings | MIT |
| | Github | preds.RData | district features used to train CRCS model | MIT |

# B  Use cases

## B.1 Structural design and water drainage network design

Here, we will describe additional details on the crash structure design and water drainage network design use cases. Both use cases were run with the same priors for the utility parameters and choice model parameters. The utility parameter prior was a uniform Dirichlet distribution. The choice model parameters for the CRCS model were chosen to capture the widest possible range of choice behaviors.

In each step, candidate choice queries were generated by sampling 1000 queries of three design options from a uniform prior over the domain of either use case. The candidate with the highest expected information gain was chosen. For crash structure design we elicited responses to 50 preference queries in each run of the experiment and for water drainage network design we elicited responses to 100 queries, though for space reasons we only show the first 75 on the graphs in this paper. The recommendations on which we measured the recommendation regret at each step were selected by maximizing the expected utility under the current posterior over a pre-calculated Pareto

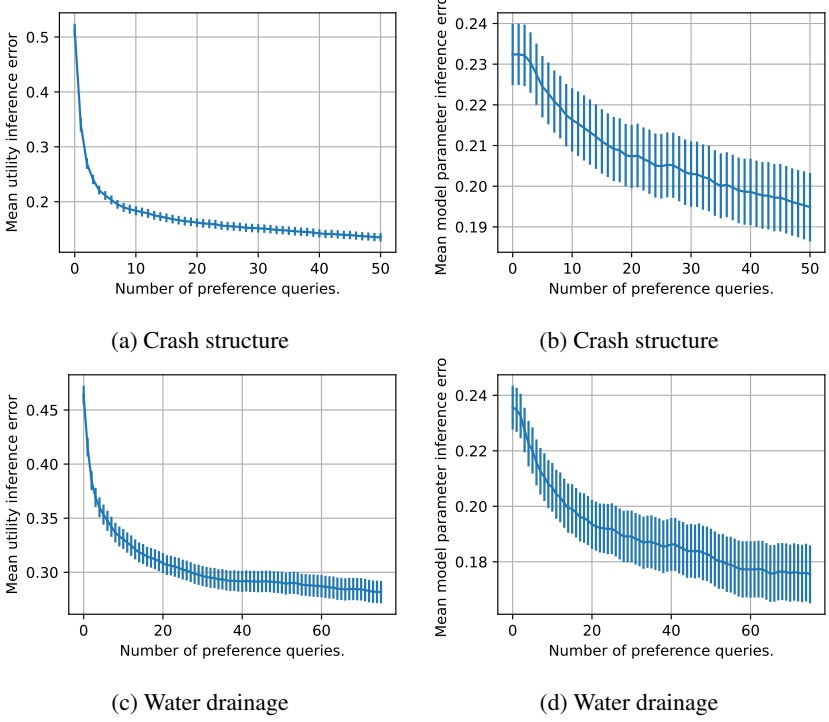

(a) Crash structure

(b) Crash structure

(c) Water drainage

(d) Water drainage

Figure 6: Additional Results for the experiments on car crash structure design and water drainage network design. (a) Utility parameter inference error on car crash structure design as a function of the number of queries put to the user. (b) Choice model parameter inference error on car crash structure design as a function of the number of queries put to the user. (c) Utility parameter inference error on water drainage network design as a function of the number of queries put to the user. (d) Choice model parameter inference error on water drainage network design as a function of the number of queries put to the user. All plots show the mean $\pm$ twice the standard error around the mean.

front for the user case. The user's optimal design was found by optimizing the true utility over this same front.

Figure 6 shows the utility and choice model parameter inference errors for both use cases.

## B.2 Retrosynthesis planning

### B.2.1 Aizynthfinder

Here we provide further details about Aizynthfinder [45]. Aizynthfinder[3] is an open-source retrosynthesis planner that uses Monte Carlo Tree Search (MCTS) and a template-based expansion policy[4] to search for possible reactions and an additional filter policy that discard the infeasible reactions. The expansion policy is a multi-class classification model that predicts the most probable reaction templates. In practice, this model produces the top 50 possible templates as the possible action during the tree expansion process. Then, the infeasible reactions are filtered out with the filter policy. During the expansion and selection phase of MCTS, it uses the upper confidence bound (UCB) to select and score routes as defined:

$$UCB = \frac{Q}{n} + C\sqrt{2\frac{\ln n - 1}{n}} \qquad (5)$$

where $n$ is the visitation times of a node, $C$ is the bias hyperparameter set to 1.4, and $Q$ is the accumulated reward that is defined as:

$$Q = 0.95 * \frac{N_{molecules\ in\ stock}}{N_{molecules}} + 0.05 \times depth \qquad (6)$$

---

[3]code available: http://www.github.com/MolecularAI/aizynthfinder

[4]download available: https://doi.org/10.6084/m9.figshare.12334577.v1

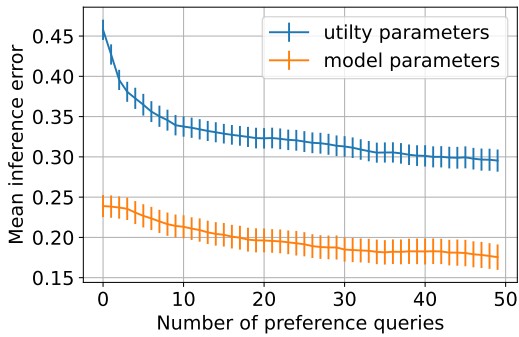

Figure 7: Results for the experiments on retrosynthesis planning. We show the mean inference error for the utility and choice model parameters as a function of the number of queries given to the user. Results are shown with the mean ± twice the standard error around the mean.

Table 7: Overview of layer sizes and training hyperparameters for $\widehat{u}$ and $\widehat{q}$ for the choice tasks considered in the experiments.

| | | embedding (3 layers) | | main (4 layers) | | | |
| | | output dims | emb. dim | output dims | batch | epochs | lr start/end |
|---|---|---|---|---|---|---|---|
| Risky Choice | $\widehat{u}$ | 128,64 | 64 | 512,256,128,3 | 1024 | 35000 | 1e-3/1e-5 |
| | $\widehat{q}$ | 128,64 | 64 | 1024,256,128,3 | 1024 | 50000 | 1e-3/1e-6 |
| Hotels | $\widehat{u}$ | 256,256,3 | 128 | 512,512,256 | 8192 | 10000 | 1e-3/1e-3 |
| | $\widehat{q}$ | 256,256 | 128 | 512,512,256,3 | 2048 | 50000 | 1e-3/1e-4 |
| District-Smart | $\widehat{u}$ | 256,256 | 128 | 1024,512,128,2 | 8192 | 20000 | 1e-2/1e-4 |
| | $\widehat{q}$ | 512,256 | 256 | 1024,1024,256,2 | 4096 | 60000 | 1e-3/1e-4 |
| Car-Alt | $\widehat{u}$ | 256,256 | 256 | 512,256,128,6 | 8192 | 30000 | 1e-3/1e-3 |
| | $\widehat{q}$ | 256,256 | 256 | 1024,1024,256,6 | 4096 | 100000 | 1e-3/1e-3 |
| Dumbalska | $\widehat{u}$ | 256,256 | 128 | 512,512,256,3 | 4096 | 40000 | 1e-3/1e-4 |
| | $\widehat{q}$ | 128,128 | 128 | 512,256,128,3 | 8192 | 25000 | 1e-3/1e-3 |
| Crash Structure | $\widehat{u}$ | 128,64 | 64 | 512,256,128,3 | 1024 | 50000 | 1e-3/1e-6 |
| | $\widehat{q}$ | 128,64 | 64 | 1024,256,128,3 | 1024 | 35000 | 1e-3/1e-5 |
| Water Drainage | $\widehat{u}$ | 128,64 | 64 | 512,256,128,3 | 1024 | 35000 | 1e-3/1e-5 |
| | $\widehat{q}$ | 128,64 | 64 | 1024,256,128,3 | 1024 | 50000 | 1e-3/1e-6 |
| Retrosynthesis | $\widehat{u}$ | 128,64 | 64 | 512,256,128,3 | 1024 | 35000 | 1e-3/1e-5 |
| | $\widehat{q}$ | 128,64 | 64 | 1024,256,128,3 | 1024 | 50000 | 1e-3/1e-6 |

where $N_{molecules\ in\ stock}$ is the current number of molecules in stock according to a given database, $N_{molecules}$ is the total number of the molecules in the current search tree and $depth$ is the depth of the search tree.

### B.2.2 Inference results

Now, we report the inference results over the preference weighs. First, we simulate the synthetic user weights by sampling 100 different weight combinations from a uniform Dirichlet distribution $\mathrm{Dir}(\alpha)$ with $\alpha = (1, 1, 1, 1, 1, 1)$. Figure 7 shows that both inference error and recommendation regret of the utility function and choice model parameters reduce during the inference using 50 preference queries.

In addition, we report the average number of solved routes from both Aizynthfinder and our model. For Aizynthfinder, we synthesize just 100 molecules, while for ours, we collect the statistics over 100 the target molecules for 100 inferred user utilities. The average number of solved routes is $45.62 \pm 28.99$ and $42.75 \pm 28.89$ for Aizynthfinder and ours respectively. These results are justified, as the synthetic user utilities are randomly sampled from non-informative prior.

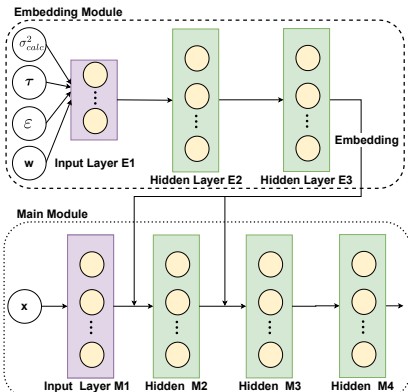

Figure 8: Overview of the network architecture of $\widehat{q}$. $\widehat{u}$ has the same architecture but takes observations $(\widetilde{\boldsymbol{u}}, \widetilde{\boldsymbol{o}})$ as input. The outputs of $\widehat{q}$ are additionally transformed by a log-softmax function (not shown).

## C  Surrogate architecture and training

We provide details here on the architecture of the two neural networks $\widehat{u}$ and $\widehat{q}$. Both networks are multi-task networks; they make their predictions conditioned on the utility parameters $w$ and the choice model parameters $\theta = (\sigma_{calc}, \varepsilon, \boldsymbol{\tau})$. $\widehat{u}$ takes as input a vector of observations $\widetilde{\boldsymbol{u}}, \widetilde{\boldsymbol{o}}$ and predicts the expected utility of each option. $\widehat{q}$ takes as input a set of options $\boldsymbol{x}$ and predicts the likelihood that each option will be chosen.

The architecture of both networks is virtually identical, differing only in their inputs, and in the fact that the output of $\widehat{q}$ is transformed with a log-softmax function, while $\widehat{u}$'s is not. Figure 8 visualizes the architecture of both networks. The networks consist of two modules. The first in an embedding module consisting of three layers that takes the utility and choice model parameters and embeds them into a latent embedding space. The second module, the main module, consists of four layers and takes as input the set of options (or the observations in the case of $\widehat{u}$) and transforms these into log-likelihood predictions (expected utilities for $\widehat{u}$). To condition the main module on the utility and choice model parameters, we concatenated the embedding from the embedding module with the input to layers 2 and 3 of the main module (see Figure 8). We trained $\widehat{u}$ and $\widehat{q}$ using the AdamW optimizer implemented in pytorch with an exponentially decaying learning rate.

Both networks have a number of hyperparameters such as layer output sizes. Other hyperparameters include training details such as learning rates and batch sizes. These hyperparameters were selected using a grid search over a predefined set of possible values for each hyperparameter. We selected the hyperparameters that minimized the loss of each surrogate neural network on each individual choice task. Table 7 lists the selected network hyperparameters: the layer output sizes for both the embedding module and main module, and the size of the embedding produced by the embedding module. The last three columns list training hyperparameters: the batch size, the number of training epochs, and the learning rate at the start and end of training.

## D  Computational resources

In table 8 we report the computational cost of the experiments reported in this paper. We only report the resources used to obtain the results presented. We estimate that if we were to include all testing and preliminary runs, the total compute time used would double or triple. We do not report the cost of validating the CRCS model on risky choice as the runtime was negligible.

All the experiments were run on either CPUs on a cluster or on a MacBook Pro. The cluster CPU jobs were single core unless otherwise mentioned. MacBook CPU jobs used multiple cores (typically less than 2.5 cores) with 16GB of memory. We estimate the maximum power consumption of a single core on our cluster to be around 7.5W, and the power usage of a single core on a MacBook to be significantly less. Assuming the worst case scenario where all computation was run on the cluster, the total energy used to obtain the reported experiments would be 146KWh. Given an average carbon

Table 8: Runtime for the various experiments reported in this paper. The column "runtime" lists the average computational resources used by each run of the experiment. The total time column lists the resources used by all runs combined.

| Use case / Dataset | Experiment | Infrastructure | Hardware | runtime | Total time |
|---|---|---|---|---|---|
| Hotel | $\widehat{u}$ training | MacBook | CPU | 1h | 1h |
| | $\widehat{q}$ training | MacBook | CPU | 1h | 1h |
| | NLL evaluation | MacBook | CPU | 7h | 7h |
| District-Smart | $\widehat{u}$ training | cluster | CPU | 15h | 15h |
| | $\widehat{q}$ training | cluster | CPU | 29h | 29h |
| | NLL evaluation | MacBook | CPU | 7h | 7h |
| Car-Alt | $\widehat{u}$ training | MacBook | CPU | 3h | 3h |
| | $\widehat{q}$ training | MacBook | CPU | 5h | 5h |
| | NLL evaluation | MacBook | CPU | 20h | 20h |
| Dumbalska | $\widehat{u}$ training | MacBook | CPU | 1h | 1h |
| | $\widehat{q}$ training | MacBook | CPU | 1h | 1h |
| | elicitation BT | MacBook | CPU | 8s | 10m |
| | elicitation BB | MacBook | CPU | 10s | 13m |
| | elicitation LCL | cluster | CPU | 28h | 2100h |
| | elicitation CRCS | cluster | CPU x2 | 20h | 1500h |
| | elicitation LC-CRCS | cluster | CPU x2 | 40h | 3000h |
| | NLL evaluation | cluster | CPU | 1h | 189h |
| Crash Structure | $\widehat{u}$ training | MacBook | CPU | 1h | 1h |
| | $\widehat{q}$ training | MacBook | CPU | 2h | 2h |
| | elicitation | cluster | CPU | 9h | 4500h |
| Water Drainage | $\widehat{u}$ training | MacBook | CPU | 2h | 2h |
| | $\widehat{q}$ training | MacBook | CPU | 5h | 5h |
| | elicitation | cluster | CPU | 10h | 3000h |
| Retrosynthesis | $\widehat{u}$ training | MacBook | CPU | 1h | 1h |
| | $\widehat{q}$ training | MacBook | CPU | 1h | 1h |
| | generation w/ AIZ | cluster | CPU | 6h | 6h |
| | generation w/ CRCS | cluster | CPU | 6h | 589h |
| | evaluation | cluster | CPU | 30m | 30m |

intensity of 94g CO2eq/KWh for the electricity supplied to our cluster in 2023, this means that we estimate the carbon footprint of these experimental results to be around 14kg CO2eq.

