# OpenReview forum: "Preference Learning of Latent Decision Utilities with a Human-like Model of Preferential Choice"
_NeurIPS.cc/2024/Conference — NeurIPS 2024 poster_

### Official Review · Reviewer_cVZM · 2024-07-08

**Soundness:** 3
**Presentation:** 3
**Contribution:** 3
**Rating:** 7
**Confidence:** 3

**Summary:**

This paper uses a state-of-the art-models model from psychology for preference learning. To make learning tractable, the authors use variational inference to approximate unknown distributions (and expectations). The model is shown to outperform existing approaches, suggesting that it better predicts human preferences and captures contextual choice effects that are commonly observed in human behaviour.

**Strengths:**

- Preference learning is a very relevant topic. This work borrows a state-of-the-art model cognitive model from psychology (Howes et al., 2016) and aims to make it computationally tractable to be applied in the context of preference learning.

- The paper is well-motivated and clearly written. It effectively balances the mathematical theory with intuitive explanations.

- I find this work to be a nice example of cross-disciplinary ML project: model from psychology, use of variational methods to make it computationally feasible, diverse set of experiments (e.g. policy making, hotel/property rankings)

- Thorough empirical evaluation: section 4.1 ensures that the learnt surrogate is a good approximation (when compared to MCMC); then in the following subsections, CRCS is evaluated on a diverse set of datasets (as already mentioned in the previous bullet), as well as simulated case studies. K-fold cross validation is performed, and Wilcoxon rank test  is done to check for statistical significance.

**Weaknesses:**

No major weaknesses. Some minor suggestions:

- Line 56 and the whole paragraph after: the meaning of CRCS acronym is not introduced until line 125.

- line 153: when reading this paragraph for the first time, I found it strange that the parameters $w$ are observed (known). It then became clear that they are observed only by the *user* that is being modelled.

- a graphical model describing all the variables in the model can be useful; it can help visualise the differences between the modelled user and the outside observer (which variables are observed or unobserved). Clearly indicating what is given and what is learnt - e.g. the prior $p(w)$ and the marginal $p(x)$ are both assumed given?

- line 221: typo though -> thought

**Questions:**

1. What is the number of parameters for each of the models? It will be good to have these reported in Table 1.
2. I might have missed it, but I didn't see the network architectures for $\hat{u}$ and $\hat{q}$?
3. Have you thought applying the model to LLMs for preference learning?

**Limitations:**

Some of the limitations of this work are discussed in Section 5. Societal impact statement is also included in Section 5.

---

> ### Author Rebuttal · Authors · 2024-08-07
>
> Thank you for the review of our paper and thoughtful feedback. We address your main questions and feedback below:
> >I find this work to be a nice example of cross-disciplinary ML project: model from psychology, use of variational methods to make it computationally feasible, diverse set of experiments.
>
> Thank you for the positive feedback!
> >What is the number of parameters for each of the models? [what are] the network architectures for $\hat{u}$ and $\hat{q}$.
>
> We will include details regarding network architectures and sizes to the appendix. We have provided a summary of these details in the general response. In the PDF attached to the general response, you can find the number of parameters for each model in Table 3, and a visualization of the network architecture in Figure 1a.
> > Have you thought [of] applying the model to LLMs for preference learning?
>
> This is a very good point. LLMs have had a tremendous impact on preference learning research and properly aligning them to human preferences is an important societal problem. Our results suggest that modeling context effects is important when learning from human preferences, and so we do consider applying our model to LLMs important future work. Right now our model assumes that options are represented by features. Thus, given some featurization of, say, LLM text outputs for a given prompt, our model could be directly applied. However, this would ignore the reading and interpreting of these texts that human evaluators have to do. We think additionally modeling those tasks in the choice model, by extending the foundation provided by the current model, could be even more transformational. Based on what we have written here, we will add a more detailed discussion on the implications of our work to LLM fine-tuning to the related work section.
> > the meaning of CRCS acronym is not introduced until line 125.
>
> Thank you for pointing this out. We will introduce the acronym earlier.
> >a graphical model describing all the variables in the model can be useful
>
> Agreed! We have added two graphical models in the PDF attached to the global response. One shows the problem from the point of view of the user making a choice (Figure 1b in the attached PDF) and the other shows it from the point of view of an outside observer trying to infer the user’s utility function (Figure 1c in the attached PDF). We will add these figures to the appendix of the paper.
> >line 153: when reading this paragraph for the first time, I found it strange that the parameters are observed (known). It then became clear that they are observed only by the user that is being modelled.
>
> Thank you for pointing this out. As another reviewer also got confused by this, we will include the graphical models and also update the writing to clarify that these observations indeed happen in the user's head and are not observable from the outside.

---

> ### Comment · Area_Chair_fW6Z · 2024-08-11
> **Please respond to the authors**
>
> Hello reviewer cVZM: The authors have responded to your comments. I would expect you to respond in kind.

---

### Official Review · Reviewer_hXMa · 2024-07-08

**Soundness:** 3
**Presentation:** 3
**Contribution:** 2
**Rating:** 4
**Confidence:** 3

**Summary:**

The paper introduces two models for learning preferences from human choice behaviors: the Computationally Rational Choice Surrogate (CRCS) and a variant (LC-CRCS) that considers the context effect of the choice set. Both models are based on a state-of-the-art cognitive model of human decision-making. The paper replaces intractable computations in the original model with surrogates to enable feasible estimations and demonstrates the numerical performance of these models through several experiments.

**Strengths:**

The paper is well-written and presents a new model, which is based on Howes et al. (2016), for understanding human choice behaviors. Compared to the original model, the new model is more computationally tractable. Also it uses several real-world datasets to test the models.

Andrew Howes, Paul A Warren, George Farmer, Wael El-Deredy, and Richard L Lewis. Why contextual preference reversals maximize expected value. Psychological review, 123(4):368, 2016.

**Weaknesses:**

1. Limited Applicability: The models require data that includes the orderings of each feature (or their generation processes) and the true utilities (or their generation processes). Such data is not commonly available. Additionally, prior knowledge of several parameters must be estimated before training the models. For instance, each feature needs a distribution for the corresponding $\tau$, which becomes challenging when the number of features is large.

2.Experimental Focus: The experimental results primarily focus on negative log-likelihoods (NLLs). Although the proposed methods outperform benchmarks in terms of NLLs, it is unclear if this improvement is practically significant. Comparing other metrics (e.g., accuracy) would better demonstrate the contribution of this work. Moreover, the paper does not provide standard deviations for the results to indicate robustness, even though the results are statistically significant.

3. Training Robustness: Line 561 mentions that "As CRCS and LC-CRCS have non-convex likelihoods, we performed gradient descent multiple times, starting from multiple starting points, and chose the parameters that achieved the best train set log-likelihood." It would be helpful to show the mean and standard deviations over several random seeds instead of just the best result to demonstrate training robustness. Training on large datasets can be time-consuming, making multiple training runs impractical.

**Questions:**

1. How can the true utility or its generation process be observed or estimated in real-world scenarios?
2. How should the prior distributions in Table 3 be selected or estimated? Are there general methods for estimating these prior distributions from a given dataset?
3. How do the models perform on other metrics (e.g., accuracy)?
4. What are the architectures of neural networks $\hat{q}$ and $\hat{u}$?

**Limitations:**

See weaknesses.

---

> ### Author Rebuttal · Authors · 2024-08-07
>
> Thank you for your thoughtful review. We address specific questions below:
> >Limited Applicability: The models require data that includes the orderings of each feature […] and the true utilities
>
> This is incorrect.The data we used was straightforward choice data consisting of the options presented to the user and the choices they made. While our cognitive model does assume that choices are informed by the ordering and utility observations you mention, these observations happen “inside the choice maker’s head” and are thus latent to an outside observer (see eq. (3)). We will update the paper to be more explicit about this. We have also included graphical models (Figure 1b and 1c in the rebuttal PDF document attached to the general comment) which clarify which variables are observed and which are not to both the choice maker and an outside observer.
> >The experimental results primarily focus on negative log-likelihoods (NLLs). […] Comparing other metrics (e.g., accuracy) would better demonstrate the contribution of this work.
>
> The evaluation of our model and the baselines on choice data sets follows prior work [18]. All models we consider predict a probability for each choice (they are probabilistic classifiers), rather than a single deterministic choice. This is important as human choices do not appear to be deterministic. Thus, our main concern is that these models should be well-calibrated. This is not something we can check with a measure like accuracy. Rather, it requires a proper scoring rule, like the NLL we used. In our other experiments - which were more representative of a typical preference learning setting - we did use other metrics to show significant practical improvement. For example, we measured how consistent a ranking implied by the inferred utility is with ranking collected from users in section 4.2.1.
> >the paper does not provide standard deviations for the results to indicate robustness, even though the results are statistically significant.
>
> We agree that this would be a good supplement to the statistical significance tests already carried out in the paper. We will add to the appendix a table listing the mean and standard deviations of the averaged negative log-likelihoods obtained over the test folds in the choice data experiments (section 4.2). A separate set of utility and choice model parameters was inferred for each test fold, so this should give readers a good idea of the robustness of the models. We have added this table to the PDF attached to the global response (Table 1).
> >[regarding L561: doing gradient descent multiple times] It would be helpful to show the mean and standard deviations over several random seeds instead of just the best result to demonstrate training robustness.
>
> We should stress that we used gradient descent multiple times to infer the latent choice model and utility parameters from human data, not to train the CRCS model itself. Even on the largest choice dataset, each inference run took only a few minutes. As requested, we will add a table to the appendices showing mean and standard deviations of the average NLL obtained over several independent gradient descent runs. We have added this table to the PDF attached to the global response (Table 2).
> >How can the true utility or its generation process be observed or estimated in real-world scenarios?
>
> This question gets to the heart of what makes human modeling difficult. As we cannot look inside people’s heads, it is impossible to observe humans’ true utilities or their true choice making process. Modeling lust follow the standard scientific process of proposing a new model based on some theory or assumptions of how this process works, and then evaluating that model against competing baselines. A choice model (including the underlying utility function) can be evaluated in two ways: i) we can check if the model correctly predicts people’s choices (experiments in section 4.2 and 4.3) and ii) we can check if the inferred utility is consistent with other observations that are derived from that same utility function (e.g., the evaluation of the implied rankings we performed in section 4.2.1).
> >How should the prior distributions in Table 3 be selected or estimated? Are there general methods for estimating these prior distributions from a given dataset?
>
> The priors for the choice model and utility parameters have very little effect on the model itself. However, as they are used when training the model, it is important that they cover any parameter values that we may wish to use in practice. We followed standard Bayesian practice for selecting them. As we had little prior knowledge of which parameter values were most likely in practice (beyond some insights from Howes et al. [17] regarding which parameter values actually affected the model’s output), we used flat priors that were as broad as sensible (see last paragraph of appendix A.1).
> >What are the architectures of neural networks $\hat{q}$ and $\hat{u}$.
>
> We will add these details to the appendix. We have provided an overview of the architecture of both networks in the general response and a visualization in Figure 1a in the accompanying PDF.

---

> > ### Comment · Reviewer_hXMa · 2024-08-12
> >
> > Thank the authors for the response. I will maintain my original score.

---

> > > ### Author Response · Authors · 2024-08-12
> > >
> > > Thank you for your response. We thought that we had fully answered your questions. To help us understand your decision, could you please point us to which of our responses failed to address your original comments? Most importantly, in answer to your first point regarding limited applicability, our model does not require observed feature orderings or true utilities. In answer to your second point, choice predictions are probabilistic requiring NLLs, not accuracy, to fully assess.

---

> > > > ### Comment · Reviewer_hXMa · 2024-08-13
> > > >
> > > > Thank you for your comment. My primary concern is still the applicability in practice: there are too many hyperparameters to decide in the proposed approach which can influence its performance, such as the priors. In your response to the choices of the priors, you mentioned, "The priors for the choice model and utility parameters have very little effect on the model itself. However, as they are used when training the model, it is important that they cover any parameter values that we may wish to use in practice." If the only factor that matters is that "they cover any parameter values that we may wish to use in practice" and "The priors for the choice model and utility parameters have very little effect on the model itself", why not always use the family of truncated normal distributions with a large range (instead of using a uniform distribution in the Hotel data or varied ranges in different datasets)? I would expect more discussions and potentially with an ablation study on the priors (and other hyperparameters), which could help demonstrate your claims and provide guidance on how to apply this approach in practice.

---

> > > > > ### Author Response · Authors · 2024-08-13
> > > > >
> > > > > Thank you for your explanation, and our apologies if we had been unclear.
> > > > >
> > > > > > My primary concern is still the applicability in practice: there are too many hyperparameters to decide in the proposed approach which can influence its performance, such as the priors.
> > > > >
> > > > > Note that the only hyperparameters that need to be chosen are the prior distributions from which samples are drawn when training the surrogate. Given that the surrogate is a highly flexible neural network, **what mainly matters here is that the priors cover all cases** which would be encountered when the surrogate is used in practice. The primary reason that there is variation in the priors we chose is that **we automatically scaled them according to the magnitude of the different features**. We could have used the same broad prior throughout but would have paid a penalty in compute time while the results of our experiments would have been identical. We believe therefore that **the priors have no negative implications for the applicability in practice** of the reported approach.
> > > > >
> > > > > To be more specific, the choice of priors is easily achieved as long as two constraints are respected: (1) priors should not be overly narrow – because this may lead to poor performance, (2) priors should not be overly broad – because this may lead to inefficient learning of the surrogate. Within these constraints there is a wide range of priors that will produce virtually identical results to those reported in the paper. The effect of the choice of priors on the performance is easy to see. Recall that our surrogate is trained to reproduce choices predicted by an existing model. We train this surrogate to be able to make these predictions conditional on a range of values of the free parameters in the model - these are the choice model and utility parameters. During training we randomly sample parameter values form the priors and then train the surrogate against the model (both conditioned on these parameters; i.e., like sampling tasks in multi-task learning). In testing, we used this surrogate to infer the values of these parameters from human data, and to make new predictions. The concern when the priors are too narrow is that we may end up in a situation where we have to make predictions with the surrogate conditioned on parameter values on which it has not been trained. This would likely result in erroneous predictions. The concern with broadening the priors is that more training is needed to cover the greater variation in parameter values to condition on. This trade-off between computational cost and accuracy across the parameter range is subjective, but was made based on prior knowledge available from prior work (as explained in our original rebuttal and in the appendix).
> > > > >
> > > > > > why not always use the family of truncated normal distributions with a large range (instead of using a uniform distribution in the Hotel data or varied ranges in different datasets)?
> > > > >
> > > > > This is essentially what we did. For Hotels, which is the first experiment we performed, we used a uniform distribution with wide support. For the other choice tasks we switched to even broader priors. There, we used the same truncated normal priors (through re-scaled due to differences in feature magnitudes) and the same Beta prior for $p(\varepsilon)$ for all tasks. We did not repeat the Hotels experiment with this choice of priors because there would have been no appreciable change in performance.
> > > > >
> > > > > **We will expand the existing discussion on prior selection in section A.1 of the appendix** to include this discussion on the factors that go into prior selection and will explain why the chosen priors differed between choice tasks.

---

> > > > > > ### Comment · Reviewer_hXMa · 2024-08-14
> > > > > >
> > > > > > Thank you for the response, it addresses some of my concerns.
> > > > > >
> > > > > > However, in my view,
> > > > > >
> > > > > > - Ablation study is important. Even the choices of priors are not important, it should be validated by experiments and to see if there are some choices can fit the tasks better and improve the performances.
> > > > > > - Besides the priors, there are still other hyperparameters such as the number of layers in the two networks. (So I don't think "Note that the only hyperparameters that need to be chosen are the prior distributions from which samples are drawn when training the surrogate.") I also do think they are important. For example, the number of layers and embedding dimensions may control the expressiveness of these two networks and thus influence the performance.

---

> > > > > > > ### Author Response · Authors · 2024-08-14
> > > > > > >
> > > > > > > Thank you for your response and for clarifying your remaining concerns.
> > > > > > >
> > > > > > > > Ablation study is important. Even the choices of priors are not important, it should be validated by experiments and to see if there are some choices can fit the tasks better and improve the performances.
> > > > > > >
> > > > > > > Yes, we naturally agree ablation studies are important when they add new insights to the paper. Our concern is that an ablation on the priors used to train the surrogate would provide a foregone conclusion. If the priors are too narrow and the surrogate is used when conditioned on parameter values it has not been trained on, the performance will be worse. In this case the prior should have been revised to be more broad. If the priors are broad, the surrogate takes longer to train.
> > > > > > >
> > > > > > > > Besides the priors, there are still other hyperparameters such as the number of layers in the two networks. (So I don't think "Note that the only hyperparameters that need to be chosen are the prior distributions from which samples are drawn when training the surrogate.") I also do think they are important. For example, the number of layers and embedding dimensions may control the expressiveness of these two networks and thus influence the performance.
> > > > > > >
> > > > > > > You are right about the network hyperparameters. We described their values in the rebuttal PDF. These were set automatically by performing a grid search over a range of possibilities. The set-up for the grid search was the same across the choice tasks. We will describe how this grid search was done in the appendix, and will provide a sensitivity analysis on the effect of the various choices of network hyperparameters on performance.
> > > > > > >
> > > > > > > Thank you again for helping improve our submission. We very much appreciated the time and effort. We believe we have been able to clarify all the concerns and would appreciate if you would take that into account in the score.

---

> ### Comment · Area_Chair_fW6Z · 2024-08-11
> **Please respond to the authors**
>
> Hello reviewer hXMa: The authors have responded to your comments. I would expect you to respond in kind.

---

### Official Review · Reviewer_rsF2 · 2024-07-12

**Soundness:** 3
**Presentation:** 3
**Contribution:** 2
**Rating:** 5
**Confidence:** 3

**Summary:**

In this paper, the authors present a new model which approximates an existing intractable Bayesian model for preference learning. The paper describes a generative model for choice, and shows how inference can be approximated using two different neural networks. The proposed model is then augmented using a cross-feature correction inspired by another piece of existing work. Results show that the proposed model outperforms the baselines on sensible metrics like negative log likelihood of the test set.

**Conclusion:** Overall, the paper makes a neat proposal and is fairly meticulous in developing the central idea. However, the contribution doesn’t appear substantial and might be of interest to a smaller subset of the community.

**Strengths:**

1. The paper provides ample background and motivating examples, clearly elucidating the relevance of the problems.
2. The development of the model is systematic, and key steps are formally defined making it easy to follow the progression of the ideas.
3. The model appears more expressive in general since it takes into account different sources of noise, and more independent variables (such as the cross-feature influence or the error tolerance of comparing features).
4. The results seem to be generally promising on both real and synthetic data.

**Weaknesses:**

1. While it was easy to follow the theory, I found it a little difficult to deeply understand the application section. A little more focus on methodology on at least one experiment might have been more helpful. Specifically, it was a little tricky to cleanly connect the theory to the application for me.
2. The contribution is somewhat incremental, and might be better suited for area-specific workshop. Since the Bayesian model already exists, the main contribution is estimating the two intractable quantities using NNs, if I understand correctly. It’s useful to have such models, but I am not sure if they generate any foundational insights.
3. For synthetic data, if the generative model is closer to the inference model, it’s not very surprising that the latter does better than another model unaware of the generative process. This weakens the evidence on synthetic data a little.

**Questions:**

NA

---

> ### Author Rebuttal · Authors · 2024-08-07
>
> Thank you for the review and feedback on our paper. We address specific questions below:
> >While it was easy to follow the theory, I found it a little difficult to deeply understand the application section. A little more focus on methodology on at least one experiment might have been more helpful. Specifically, it was a little tricky to cleanly connect the theory to the application for me.
>
> Thank you for this feedback. We will update the paper to more explicitly connect the data available in sections 4.2 and 4.3 to the observed and unobserved variables discussed in section 3: In 4.2 we are given a large dataset of tuples $(x^{l},y^{l})$ where each $x^{l}$ is a set of options presented to a user and each corresponding $y^{l}$ is the observed choice. Given this data we now look to infer the unobserved utility parameters $w$ and choice model parameters (e.g. $\varepsilon, \boldsymbol{\tau}, \sigma^2_{calc}$ in the case of CRCS) that best explain this data. The difference in 4.3 is that the dataset is not given all at once, rather we have access to the option sets $x^{l}$ and can actively select which $y^{l})$ we want to observe.
> >The contribution is somewhat incremental [...] Since the Bayesian model already exists, the main contribution is estimating the two intractable quantities using NNs, [...] It’s useful to have such models, but I am not sure if they generate any foundational insights.
>
> We disagree that the contribution is incremental. The foundational insight is that a general theory of human cognition - namely computational rationality - offers a powerful and general inductive bias for machine learning. We study how to mitigate the effect context effects - a human bias widely studied in cognitive science - have on utility inference from preferences. Whereas prior work has attempted to account for these effects by learning them from data without any such inductive biases [18,35], we follow a model-based approach which leverages an existing cognitive model introduced by Howes et al. [17]. This model, which is an instantiation of computational rationality for decision making, is empirically validated to reproduce known context effects. This provides us with strong inductive biases that significantly improve utility function inference and choice prediction compared to prior work, showing that our model-based approach works for the important problem of learning from preferences. But the cognitive science literature suggests that deriving inductive biases from computational rationality can generalize to any setting in which people make decisions [A,B,C]. This is in our view a foundational contribution to both cognitive science and machine learning. We will discuss this contribution more explicitly in the introduction and discussion of the paper.
>
> We also count a number of other tangible contributions. We have introduced a new choice model, CRCS, which improves utility function inference in preference learning compared to prior work. This required making the model introduced by Howes et al. [17] tractable using surrogates. We have also provided further empirical evidence for this specific model. This is significant because SOTA cognitive models such as [17, 38] are seldom tested on large choice, mainly because inference is generally intractable. Lastly, we have shown that there are context effects that the original Howes et al. [17] model does not capture, and have introduced LC-CRCS which introduced a learnable mechanism that can fit to these additional context effects.
>
> [A] Richard L. L., Howes A., and Singh S. "Computational rationality: Linking mechanism and behavior through bounded utility maximization." Topics in cognitive science 6.2 (2014): 279-311.
>
> [B] Lieder F., and Griffiths TL. "Resource-rational analysis: Understanding human cognition as the optimal use of limited computational resources." Behavioral and brain sciences 43 (2020): e1.
>
> [C] Oulasvirta A., Jokinen JPP., and Howes A. "Computational rationality as a theory of interaction." Proceedings of the 2022 CHI Conference on Human Factors in Computing Systems. 2022.
> >[The paper] might be of interest to a smaller subset of the community [...] might be better suited for area-specific workshop.
>
> We disagree. Learning from preferences is an increasingly important approach in human-in-the-loop machine learning for inferring unknown utility functions. The modeling of those preferences (choices) is therefore relevant to a wide part of the ML community. We see this reflected in the fact that earlier work on choice modeling, which we improve on, has been published at top-level ML conferences such as ICML [35] and KDD [18]. Furthermore, our work fits within a wider body of recent work published at NeurIPS on human behavior modeling [D,E,F,G].
>
> [D] Belousov B., et al. "Catching heuristics are optimal control policies." Advances in neural information processing systems 29 (2016).
>
> [E] Binz M., and Schulz E. "Modeling human exploration through resource-rational reinforcement learning." Advances in neural information processing systems 35 (2022).
>
> [F] Chandra, K., et al. "Inferring the future by imagining the past." Advances in Neural Information Processing Systems 36 (2023).
>
> [G] Teng T., Kevin L., and Hang Z. "Bounded rationality in structured density estimation." Advances in Neural Information Processing Systems 36 (2024).
> >For synthetic data, if the generative model is closer to the inference model, it’s not very surprising that the latter does better than another model unaware of the generative process. This weakens the evidence on synthetic data a little.
>
> We do not agree that this weakens the evidence. It is clear that aligning a system to a user’s utility will make the system more useful. However, how to achieve such alignment in practice is not trivial. The simulated use cases we present point to the fact that preference learning with our model is a practical solution for this.

---

> > ### Comment · Reviewer_rsF2 · 2024-08-13
> > **Update after authors' rebuttal**
> >
> > I appreciate the authors taking the time to address my concerns. The clarification around how the theory connects to the experiments is quite helpful. I also concede the point that this could be an apt addition to the main conference given the recent trend of publications the authors pointed out.
> >
> > However, I do not quite understand the assertion, `We study how to mitigate the effect context effects - a human bias widely studied in cognitive science - have on utility inference from preferences`. My understanding is context effects are widely observed and the proposed model is learning underlying utilities that can explain their emergence. This makes me wonder if I missed something fundamental in the paper. I am raising my score from 4 to 5, but lowering my confidence from 4 to 3.

---

> > > ### Author Response · Authors · 2024-08-14
> > >
> > > Thank you for asking, and for the comments.
> > >
> > > > I do not quite understand the assertion, `We study how to mitigate the effect context effects - a human bias widely studied in cognitive science - have on utility inference from preferences.` My understanding is context effects are widely observed and the proposed model is learning underlying utilities that can explain their emergence.
> > >
> > > Your understanding is correct. Context effects are widely observed and the underlying utilities are a fundamental factor in their emergence. The point we were trying to make is that when inferring these underlying utilities from choices made by humans - where we expect context effects to have affected these choices - it is important that we do so with a model that correctly models context effects. If context effects are not properly modeled, there is a risk that we infer the wrong underlying utility function. The experiments provide evidence for this. The Bradley-Terry model, which does not model any context effects, is in all cases the worst of the choice models considered.
> > >
> > > We hope this helps to clarify.

---

> ### Comment · Area_Chair_fW6Z · 2024-08-11
> **Please respond to reviewers**
>
> Hello reviewer rsF2: The authors have responded to your comments. I would expect you to respond in kind.

---

### Official Review · Reviewer_CwtL · 2024-07-13

**Soundness:** 2
**Presentation:** 3
**Contribution:** 3
**Rating:** 4
**Confidence:** 4

**Summary:**

The submission proposes an approach to preference learning using a model inspired by findings in cognitive science. Specifically, it uses an amortized inference variant of a previously proposed model to enable tractable inference of preference values, and applies it to a number of case studies, where it is shown to outperform both the classical Bradley-Terry model, and additional more recent baselines.

**Strengths:**

The paper sets up its problem well and provides a suitable review of past work. I particularly appreciated the very crisp distinction between user's and outside observer's model in section 3.2. Performance is competitive relative to the baselines chosen, which include relatively recent methods.

**Weaknesses:**

My primary concerns with the paper are related to the actual technical implementation details, which are not articulated clearly enough, and where they are articulated give me some pause. Specifically:
* If I understand things correctly, the policy network takes in $x, w, \tilde{u} and \tilde{o}$, and a sufficiently flexible surrogate should be able to learn arbitrary functions of x and w already. Is this correct? If so, why is the explicit feature engineering needed? The paper also discusses later "built in" context effects in CRCS and "learned" ones in LR-CRCS -- what is meant by this?
* This also makes me wonder: what's the network architecture used and other training hyperparameters (optimizer, learning rates, etc)? In general I would expect a sufficiently flexible surrogate to also achieve better than ~92% agreement with the original model. I'm also puzzled about the claim of insufficient option data to claim the utility network -- shouldn't it be possible to simulate arbitrary amounts of observations in this case such that the network fully spans the range of possible outcomes and enabling amortized inference? I can see why sampling only from the training data would create this restriction, but I don't understand why this choice was made.

**Questions:**

* Reporting summed rather than averaged NLLs (with standard errors) seems like an odd choice -- why was this done?
* Can the authors comment about the choice of Wilcoxon signed-rank test for some hypothesis tests and t-tests for others?
* Why are baselines not present in the regret panels of figure 1?

Additional Notes:
* Figure 1 tick labels should be larger.
* L53 maybe "computational rationality analysis"?
* L66 it's -> its
* The notation in section 3.3. is a bit confusing, in the following way: u is a function of x, but the notation sometimes uses u and sometimes x (e.g. in the expectations in the utility and policy loss, or how \tilde{u} is conditioned on u but \tilde{o} is conditioned on x. I think this might be because of a distinction between the observer's and user's likelihoods, but this isn't quite spelled out here.
* I believe that the overall setup here is a bilevel optimization problem (because L_util is optimized w.r.t. w and L_pol w.r.t. x), and I'm not sure how incomplete optimization or other issues in the utility network impact the policy network (especially considering that the accuracy of the utility network seems not ideal).
* Line 278 I believe \citet should be \citep there?

**Limitations:**

Discussion is adequate, though as noted above I'm puzzled about the limitation on requiring sufficiently many choice sets for estimation.

---

> ### Author Rebuttal · Authors · 2024-08-07
>
> Thank you for your review of our paper and your feedback. We have addressed the main questions and concerns you have raised below.
> >the policy network takes in $x, w, \tilde{u}$ and $\tilde{o}$, and a sufficiently flexible surrogate should be able to learn arbitrary functions of x and w already. Why is the explicit feature engineering needed? [what is meant by] "built in" context effects in CRCS and "learned" ones in [LC]-CRCS.
>
> Thank you for pointing out that this was unclear. To clarify, the observations $\tilde{u}$ and $\tilde{o}$ are not engineered features, but rather a core part of our cognitive model. As explained in section 3.1, the cognitive model theorizes that context effects come from the fact that humans make choices that maximize option utilities estimated from noisy observations [17], specifically noisy observations of each option’s utility ($\tilde{u}$) and noisy feature value comparisons across options ($\tilde{o}$). It has been empirically validated that maximizing these utility estimates given these observations leads to the context effects exhibited by humans. Our tractable surrogate for this model, CRCS, inherits these same context effects, which is why we describe them as “built in”. However, we found that this model did not reproduce all context effects present in the considered choice data. LC-CRCS addresses this by introducing a learnable component that is able to learn context effects not yet captured by CRCS itself. The advantage of having a model where these context effects are “built in” is that this creates inductive biases that lead to better utility function inference and choice prediction, as we show in the experiments.
> >This also makes me wonder: what's the network architecture used and other training hyperparameters (optimizer, learning rates, etc)?
>
> Although these details are available in the code we submitted in the supplement, we do agree that it would be easier for readers if they were available as part of the appendices as well. We will add details on the network architecture and sizes to the Appendix. We have provided a summary in the general response.
> >In general I would expect a sufficiently flexible surrogate to also achieve better than ~92% agreement with the original model.
>
> We used the most stringent possible definition of agreement, namely that both models had to agree on the full ordering of the three option utilities. This is clearly a high bar. Moreover, the original model we compared to uses a Monte-Carlo estimator that estimates the expected values of the option utilities from a large but finite number of samples. Thus, the quantity we compare to is not some ground truth reference but rather another (noisy) estimate. As any error could be due to either the proposed surrogate or the original model being wrong, even a perfect surrogate would likely not reach 100% agreement.
> >I'm also puzzled about the claim of insufficient option data to [train] the utility network -- shouldn't it be possible to simulate arbitrary amounts of observations[...]? I can see why sampling only from the training data would create this restriction, but I don't understand why this choice was made.
>
> Unfortunately, in this specific instance, there was insufficient detail available regarding how the original user study, reported in [37], was done; no details were available on car and participant features that were used to construct the original option sets. We were therefore unable to generate option sets from a distribution that aligned well with the original experimental condition, forcing us to sample them from the training data only. This was not an issue for the other choice tasks, where the details of the user studies were more fully documented and we were able to simulate an arbitrary number of choices. Appendix A.1 discusses these issues in more depth.
> > Reporting summed rather than averaged NLLs (with standard errors) seems like an odd choice -- why was this done?
>
> We reported summed NLLs in the same way as was done in prior choice modeling work [18]. However, we agree that additionally reporting standard deviations or errors would add value to the paper, especially - as reviewer hXMa explained - to help readers understand the robustness of the models to variations in the training data. We will add to the appendix a table listing the mean and standard deviations of the averaged NLLs obtained over the test folds in the choice data experiments (section 4.2). You can find this information in table 1 in the PDF attached to the global response.
> > Can the authors comment about the choice of Wilcoxon signed-rank test for some hypothesis tests and t-tests for others?
>
> In most of our experiments we compared different model scores (e.g. NLL) on the same static data, and thus we used paired two-sample tests to compare the scores between models. As normality generally did not hold, we used Wilcoxon signed-rank tests for this. For experiment 4.3, where we used active learning, the evaluation data differed between the models as each would have selected different points to train on, thus we used an unpaired two-sample test, in this case an independent t-test.
> We have noticed now that we misreported the hypothesis tests used in sections 4.2.1 and 4.3 – these should be switched. We used an independent t-test in 4.3 and a Wilcoxon signed rank test in 4.2.1. We will fix this in the paper and will explain briefly why each hypothesis test was chosen.
> > Why are baselines not present in the regret panels of figure 1?
>
> Comparing to baselines was not the focus of these experiments. These are synthetic tests where we measured parameter recovery and practicality of CRCS in real-world experiments. No human data was used, and instead choices were generated from our own CRCS model. Clearly, our own model will fit best to choices generated by it, so any comparison to the baselines seemed to us as though it would be disingenuous, in this instance.

---

> > ### Comment · Reviewer_CwtL · 2024-08-12
> > **Still confused about CRCS vs LC-CRCS**
> >
> > I appreciate the authors' clarifications regarding the training setup. I'm still surprised that the agreement to the brute-force approach is so low, but I will also grant the point that maybe it's not that important to match to the brute force approach considering performance of the proposed model on the actual evaluation tasks is still improved.
> >
> > Regarding feature engineering, my concern is not $\tilde{u}$ and $\tilde{o}$. Rather, I mean the $g(w, x)$ learned linear mapping -- why is this not already learnable within the original CRCS? As far as I can tell, $g$ simply takes the average of the $x$'s, and transforms it linearly, both of which the $\hat{q}$ network can do already.

---

> ### Comment · Area_Chair_fW6Z · 2024-08-11
> **Please respond to the authors**
>
> Hello reviewer CwtL: The authors have responded to your comments. I would expect you to respond in kind.

---

> ### Author Response · Authors · 2024-08-13
>
> Thank you for your comment. We apologize for misunderstanding your original question regarding LC-CRCS.
>
> Fundamentally, CRCS as we defined it does not learn context effects from (human) data. Instead, context effects emerge given an empirically validated theory (choices maximise expected utility given noisy observations; section 3.1). This theory can be simulated and thus be amortised into \hat{q}. $\hat{q}$ can therefore be trained without any access to human data. There are a number of free variables in the model (the choice model and utility parameters) which are inferred from human data in our experiments, but these determine the utility function and various noise levels within the choice process, and are thus not directly responsible for the context effects. Although not learning context effects from human data has clear advantages, it also means that the model will not generate context effects not explained by the theory. This is why we introduced the additional linear mapping $g(w,x)$ in LC-CRCS. This can be learnt from human data and can therefore fit to context effects not yet explained by the theory or generated by CRCS.

---

### Author Rebuttal · Authors · 2024-08-07

We thank all reviewers for the time and effort dedicated to review of our work and for the helpful and constructive feedback.

## Motivation and focus of the paper
Our paper presents a cross-disciplinary approach to preference learning. Humans are known to exhibit a number of context effects when making choices. When these effects are not properly accounted for, learning from choices (preferences) may lead to incorrect inferences of the underlying utility. This has important implications for ML systems that learn from preferences.

The foundational contribution of our paper is the insight that computational rationality, which posits that humans are rational under bounds, can offer a step change in how ML systems learn about humans. Concretely, we propose a model-based approach to preference learning which leverages a SOTA cognitive model [17] derived from this computational rationality theory. The empirically validated computational rationality assumptions built into this model induce known context effects. Using this model, or rather a new tractable variant of it, for learning from preferences therefore introduces a strong inductive bias in our inferences. This type of inductive bias has not been used in preference learning before, and we show that it significantly improves utility inference and choice prediction compared to prior work.

The paper therefore offers a number of concrete contributions. We introduce CRCS, a tractable version of an existing cognitive model by Howes et al. [17] that supports efficient inference of the utility function and model parameters. We show that CRCS can be used in large-scale learning from preferences and that it makes better inferences than a set of recent baselines. This has practical benefits for any existing work that uses learning from preferences. To account for any context effects not yet captured by CRCS, we additionally introduce LC-CRCS, which is able to learn any additional context effects from human data.

## Clarification on neural network architecture
As requested by several reviewers, we provide here details on the architecture of the two neural networks we use in the paper, $\hat{u}$ and $\hat{q}$. These approximate key intractable computations in the paper: $\hat{u}$ takes as input a vector of observations $(\boldsymbol{\widetilde{u}}, \boldsymbol{\widetilde{o}})$ and predicts the expected utility of each option. $\hat{q}$ takes as input a set of options and predicts the likelihood that each option will be chosen. Both networks are multitask networks; they are additionally conditioned on the parameters of the utility function $w$ and the choice model parameters $(\varepsilon, \boldsymbol{\tau}, \sigma_{calc}^2)$.

The architecture of both networks is virtually identical, differing only in the dimension of their inputs, and in the fact that the output of $\hat{q}$ is transformed with a log-softmax function. Figure 1a in the attached PDF shows the architecture of $\hat{q}$. The network design consists of two modules. The first in an embedding module consisting of three layers that takes the utility and choice model parameters and embeds them into a latent embedding space. The output dimension of the layers and the dimension of the embedding space are given in Table 3 in the attached PDF. The second module, the main module, consists of four layers and takes as input the set of options $x$ (or the observations in the case of $\hat{u}$) and transforms these into likelihood predictions (expected utilities for $\hat{u}$). The output sizes of these layers are also listed in Table 3. To condition the main module on the utility and choice model parameters, we concatenated the embedding from the embedding module with the input to layers 2 and 3 of the main module (see figure). We trained $\hat{u}$ and $\hat{q}$ using the AdamW optimizer implemented in pytorch. We used an exponentially decaying learning rate. We list the batch size, number of epochs for training and the starting and ending learning rates in Table 3 in the attached PDF.

## Additional experimental results and clarity improvements
We have collected additional empirical results to support our response to reviewers’ comments. The new figures and tables are in the rebuttal PDF document (attached to this comment). Based on the feedback from the reviewers, we will also expand the paper and incorporate a number of clarifications. We list the most important changes below:
- We will clarify the description of the model. Specifically, we will be more explicit about which variables are observable to the user and to an outside observer. We will also more clearly delineate what part of the theory concerns the cognitive model and what part concerns learning from preferences.
- As requested by reviewer cVZM we have added to the appendix two graphical models, one showing the cognitive choice model (Figure 1b in the attached PDF) and one showing how this fits within the larger inference problem of learning from preferences (Figure 1c in the attached PDF).
- We will update the paper’s introduction and discussion sections to clearly state the foundational insights contributed by this work, based on the response we gave to reviewer rsF2.
- As requested by reviewers CwtL, hXMa and cVZM we have provided further details about the architecture and training details of the neural networks above. We will add these details to the appendix of the paper.
- As requested by reviewers hXMa and CwtL we have additionally reported averaged NLL figures with standard deviations for the choice set experiments in Table 1 in the attached PDF. These will be added to the appendix of the paper.
- As requested by reviewer hMXa we have included a table showing the mean and standard deviation of the average NLL obtained over several gradient descent runs in Table 2 in the attached PDF. We will discuss this table in the appendices.

---

### Decision · Program_Chairs · 2024-09-25

**Decision:**

Accept (poster)

**Comment:**

The paper proposes modeling of choice behavior by using models of human choice from cognitive science as priors linked to data by neural network models. This allows fitting a broader set of possible data than is possible under intractable cognitive science models, and fitting to data that are not well explained by such models. Detailed experimental simulations demonstrate the efficacy of the solution.

Opinions of this paper are quite split. I have read the reviews, responses, and the paper itself. I lean toward acceptance. The reviewers had a large number of legitimate questions, which seem to be answered in the responses. Addressing these will require non-trivial work on the authors' part. The remaining question is about the contribution. The authors argue that using cognitive models as a prior is in important contribution, one that is supported by their empirical results. The authors are generally less convinced that the idea is important, given that there are not significant technical advances. I lean toward acceptance because ideas that supported by data can be important contributions in general, and in this specific case, improving modeling human choice behavior at scale is an important problem.